# Therapeutic Potential of the β3-Adrenergic Receptor and Its Ligands in Cardiovascular Diseases

**DOI:** 10.3390/ijms262411844

**Published:** 2025-12-08

**Authors:** Marcel Kempiński, Paweł Jańczak, Adrianna Porębska, Patrycja Sandra Zawadzka, Paulina Jastrzębska, Marcin Mateusz Granat, Małgorzata Wojciechowska

**Affiliations:** 1Chair and Department of Experimental and Clinical Physiology, Laboratory of Centre for Preclinical Research, Medical University of Warsaw, Banacha 1b, 02-097 Warsaw, Poland; marckemp2002@gmail.com (M.K.); pawel49j@gmail.com (P.J.); s086195@student.wum.edu.pl (A.P.); s090641@student.wum.edu.pl (P.S.Z.); pjastrzebska48@gmail.com (P.J.); 2Department of Clinical and Experimental Pharmacology, Faculty of Medicine, Medical University of Warsaw, Banacha 1b, 02-097 Warsaw, Poland; marcin.granat@wum.edu.pl

**Keywords:** β3-adrenergic receptor, adrenergic receptors, heart, blood vessels, hypertension, arrhythmias, agonists, antagonists, heart failure, atherosclerosis

## Abstract

The β3-adrenergic receptor is an exceptional member of the β-adrenergic receptor family. Recent findings suggest that it may play a more significant role in the cardiovascular system than previously recognized, partly through its potential anti-inflammatory actions. For this reason, it has been extensively studied as a potential target in cardiovascular therapy. In this review, we focus on the β3-adrenergic receptor agonists and their pharmacodynamics regarding receptor binding and activation. We then describe the mechanisms of action mediated by the β3-adrenergic receptor in blood vessels and in the heart under both physiological and pathological conditions. We particularly concentrate on the therapeutic potential in the arterial, pulmonary, and portal hypertension as well as cardiac arrhythmias, ischemia, and heart failure, considering both receptor agonism and antagonism.

## 1. Introduction

Adrenergic receptors (ARs) are members of the G-protein-coupled receptors which are composed of a single polypeptide chain that spans the membrane seven times. They are primarily activated by the catecholamines norepinephrine and epinephrine [1]. The sympathetic receptors are divided into two main classes, α and β, which are further subdivided into three α1 subtypes and three α2 subtypes, whereas the β receptors include β1, β2, and β3 [1]. G-proteins consist of three subunits (α, β, and γ) and are classified into families: G(i), G(s), G(12/13), and G(q) [2]. The β3-adrenergic receptor (β3-AR) couples to two of these families, depending on the splice variant of the receptor. β3a-AR couples only to G(s), leading to the adenylyl cyclase stimulation. In contrast, β3b-AR interacts with either G(s) or G(i), leading to the activation or inhibition of adenylyl cyclase, respectively. Moreover, G(i) stimulation can lead to an indirect increase in guanosine 3′,5′-cyclic monophosphate (cGMP) production by reducing protein kinase A (PKA) activity and increasing nitric monoxide synthase (NOS) activity [3,4,5]. The β3-AR was identified as a new subtype of the adrenergic receptor family, following the cloning of its gene from a human genomic library [6]. The human β3-AR gene is found on chromosome 8 [7].

The β3-AR is expressed in many tissues, including adipose tissue, the urinary bladder, the myocardium, and the arterial endothelium, and plays an important role in regulating various physiological functions. In the adipose tissue, β3-AR was initially shown to mediate lipolysis and to stimulate the expression of uncoupling protein 1 (UCP1), which is involved in thermogenesis. In the urinary bladder, its activation relaxes the detrusor muscle. In the myocardium and the arterial endothelium, its agonists may have a potential role in the treatment of heart failure (HF) and other cardiovascular disorders [8,9,10]. Indeed, their activation promotes vasodilation and inhibits smooth muscle cell proliferation, suggesting a protective role against vascular remodelling. On the other hand, nitric monoxide (NO) produced by endocardial cells has been shown to decrease cardiac contractility [11,12].

The widespread tissue distribution of β3-AR suggests its potential as a therapeutic target in multiple clinical areas. It has been considered for the treatment of obesity, type 2 diabetes, metabolic syndrome, preterm labour, and colon cancer [13]. This receptor serves as a therapeutic target for the management of overactive bladder syndrome [14,15], with agonists such as mirabegron or vibegron already approved for clinical use [16]. β3-AR agonist efficiency is cell-dependent; based on the location of the cell and its type, the receptor may or may not undergo desensitization [17,18]. Regarding mechanisms that may limit agonist efficacy, the activation of PKA downstream of β3-AR-induced cAMP elevation plays a dual role: it supports downstream signalling but may also regulate the receptor itself—promoting receptor sensitization via cAMP response elements in the gene, while in some settings contributing to desensitization. Desensitization can occur over time through a reduction in receptor numbers or by internalization. Repeated or long-term agonist exposure can lead to desensitization, but β3-ARs are relatively resistant in comparison with receptors like β2-ARs. This is mainly because β3-AR lacks several serine/threonine phosphorylation sites in its C-terminal tail and in the third intracellular loop, which are critical for desensitization. Directly related to this structural feature, G-protein-coupled receptor kinases (GRKs, the family including β-AR kinase) do not significantly phosphorylate the β3-AR upon agonist stimulation, unlike β2-AR. That is why, in heart failure, the expression of β3-ARs is maintained or even upregulated, supporting the hypothesis that β3-AR agonists could provide therapeutic benefits in these patients [19]. In contrast to β1 and β2-AR agonists, which typically downregulate their respective receptors, some studies report that β3-AR agonists, such as BRL37344, increase β3-AR expression. Conversely, some β3-AR antagonists have been shown to perform a downregulating effect on the receptor [10]. However, based on the literature, ligand-induced regulation of β3-AR expression is more complex, showing substantial variability depending on the cell or animal model used [19].

This paper reviews the current literature on β3-AR in the cardiovascular system and its selective agonists, focusing on their binding characteristics, receptor-specific mechanisms of action, and potential therapeutic applications in cardiovascular diseases. We conducted a comprehensive search on databases such as PubMed (accessed from 15 to 31 October 2024) and IUPHAR/BPS (accessed from 16 to 22 January 2025) in search of β3-AR ligands and their effects on the cardiovascular system. The search phrases included “β3 adrenergic receptor” and “beta 3 adrenergic receptor” combined with “agonist”, “antagonist”, “inhibitor”, “heart failure”, “cardiac arrhythmias”, “atherosclerosis”, “myocardial ischemia”, “acute coronary syndrome”, “coronary artery disease”, “angina pectoris”, “myocardial infarction”, “ST-elevation myocardial infarction”, “non-ST elevation myocardial infarction”, “ischemia-reperfusion injury”, “reperfusion injury”, and “hypertension”. The considered publications spanned from 1986 to 2024. Only studies that were fully accessible and published in English were included. The criterion for including ligands in the study depended on the presence of studies on their effects on the cardiovascular system.

## 2. β3-Adrenergic Receptor Pharmacology

### 2.1. General Information

As with all β-adrenergic receptors, the properties of β3-AR can be characterized in terms of agonist potency and efficacy, which depend on both the agonist’s characteristics (such as affinity and intrinsic efficacy) and the specific features of the target cell or tissue that determine the response [20]. Table 1 presents the determinants of efficacy, potency, and selectivity towards the human β3-AR for the agonists discussed later in this article, along with additional compounds included based on the applied methodology. We focused on cAMP accumulation in Chinese Hamster Ovary (CHO) cells as an agonist response because, to our knowledge, it is the most studied model regarding human β3-AR pharmacology and its comparison with β1 and β2-ARs. In Table 1, efficacy is expressed as the percentage of the response generated by isoproterenol; potency is represented by the half-maximal effective concentration (EC_50_), defined as the concentration of an agonist that produces 50% of its maximal effect [20]; and selectivity is indicated by the ratio of EC_50_ values across different β-AR subtypes.

As an adrenergic receptor, β3-AR is activated by endogenous catecholamines epinephrine and norepinephrine, which are non-selective β-AR agonists. While both of them show similar affinity and potency towards β1-AR, epinephrine is a stronger agonist than norepinephrine for the β2-AR. For the β3-AR, the situation is different as norepinephrine presents higher affinity and potency towards the receptor [21].

**Table 1 ijms-26-11844-t001:** Potency, efficacy, and selectivity of chosen β3-AR agonists. Given values refer to cAMP accumulation as a measured response in Chinese Hamster Ovary (CHO) cell lines. The maximal agonist effect is related to the response of isoproterenol. * these values were calculated based on the referenced article. ^a^ particle is used as an antagonist of the receptor. ND—no data. SEM—standard error of the mean.

Agonist	Efficacy			Potency			Selectivity		Isoproterenol Concentration [M]	References
	β1-AR Maximal Agonist Effect ± SEM [%]	β2-AR Maximal Agonist Effect ± SEM [%]	β3-AR Maximal Agonist Effect ± SEM [%]	β1-AR EC_50_ [nM]	β2-AR EC_50_ [nM]	β3-AR EC_50_ [nM]	EC_50_ Ratio β1-AR/ β3-AR	EC_50_ Ratio β2-AR/ β3-AR		
Mirabegron	10	10	80	>10,000	>10,000	22.4	>446	>446	10^−4^	[22]
Solabegron	4.2 ± 1.7	8.8 ± 2.4	89.1 ± 4.2	<10,000 *	<10,000 *	6.92 *	<1445 *	<1445 *	10^−5^	[23]
Vibegron	5	7	84	>10,000	>10,000	1	>10,000 *	>10,000 *	10^−6^	[24]
BRL37344	50	70	60	12,900	360	457	28.2	0.79	10^−4^	[22]
	55.7 ± 3.7	50.7 ± 6.5	46.7 ± 6	986 ± 108	153 ± 21	728 ± 203	70.1	10.1	10^−5^	[25]
	100.7 ± 8.6	80.1 ± 4.1	79.7 ± 3.6	288.4 *	131.8 *	33.9 *	8.51 *	3.89 *	10^−5^	[21]
CL316243	ND	ND	54.9 ± 2.1	>10,000	>10,000	11,200 ± 4000	>46.2	>43	10^−5^	[25]
	0	10	50	>10,000	>10,000	4430	>2.26 *	>2.26 *	10^−4^	[22]
Zinterol	113.5 ± 3.8	105.3 ± 4.6	101.2 ± 3.4	58.9 *	0.3 *	8.1 *	7.27 *	0.04 *	10^−5^	[21]
L755507	101.6 ± 3	3 ± 0.2	101.1 ± 3.4	23.4 *	89.1 *	0.1 *	234 *	891 *	10^−5^	[21]
Nebivolol	2.8 ± 0.3 ^a^	ND	ND	1.1 *^a^	ND	ND	ND	ND	10^−5^	[21]
ICI215001	ND	ND	60.3 ± 2.9	ND	ND	87.1 *	ND	ND	10^−5^	[26]
CGP12177	ND ^a^	ND ^a^	61.4 ± 1.5	ND ^a^	ND ^a^	269.2 *	ND	ND	10^−5^	[26]
ZD7114	72.0 ± 2.9	2.2 ± 0.4	58.2 ± 2	8.9 *	22.4 *	28.2 *	0.32 *	0.79 *	10^−5^	[21]
Carazolol	38.1 ± 4.9	1.9 ± 0.3	75.7 ± 1.1	40.7 *	0.2 *	109.6 *	0.37 *	0 *	10^−5^	[21]
Fenoterol	106 ± 1.7	100.9 ± 2.7	100.5 ± 4	29.5 *	1.3 *	23.4 *	1.26 *	0.06 *	10^−5^	[21]

Since the discovery of β3-AR, many substances affecting this receptor have been studied. The selectivity of these compounds differs, as the compounds may also affect β1 and β2-AR (Table 1) [27]. Therefore, many researchers decide to use β1/2-AR inhibitors, such as propranolol or nadolol, which eliminate unwanted effects from these receptors and increase the credibility of an experiment [28]. Furthermore, β-AR reserve exists in the human heart. This phenomenon occurs when an agonist triggers a bigger effect than the number of receptors it stimulates in the tissue. For example, half of the maximal agonist effect appears during occupation of less than 50% of the available receptors [29]. Consequently, higher doses of potentially selective β3-AR agonists might elicit an unexpectedly large, unwanted effect on other β-ARs [12]. Fortunately, agonists currently used in clinics (mirabegron and vibegron) present relatively high selectivity towards β3-AR (Table 1).

### 2.2. Gene Polymorphisms of β3-AR and Their Influence on Its Pharmacology

Several polymorphisms of the β3-AR gene have been identified, which may contribute to phenotypic variability. Some of these variants have been associated with conditions such as obesity and type 2 diabetes mellitus [30], coronary artery disease with potentially protective effects [31], and overactive bladder syndrome [32]. However, findings regarding their functional significance remain inconsistent, highlighting the need for further investigation. Of particular interest is the potential impact of these polymorphisms on drug response. For example, in Japanese subjects, β3-AR gene polymorphism *ADRB3*: T727C-Trp64Arg has been positively correlated with the antihypertensive effect of thiazide therapy [33]. Nevertheless, Vrydag et al. examined several gene polymorphisms in human embryonic kidney 293 (HEK293) cells stimulated with five agonists (including noradrenaline, isoproterenol, and mirabegron) and found no significant effect on cAMP accumulation. However, because HEK293 cells lack some signalling pathways (e.g., NOS), the study could not assess certain downstream mechanisms [34].

### 2.3. Ligand-Directed Agonism

β3-AR stimulation exerts cardiovascular effects opposite to those of β1- and β2-ARs [35]. Its activation requires higher concentrations of catecholamines [36]. Additionally, due to its resistance to desensitization, it may act as an endogenous “β-blocker”, offering protection against catecholamine overstimulation commonly seen in cardiovascular pathologies [19,37]. A phenomenon known as ligand-directed agonism (also called biassed agonism) was observed for β3-AR [38]. Agonists are referred to as “biassed” when they have the ability to activate only a specific signalling pathway rather than all possible pathways upon binding to the receptor [39]. Sometimes, one ligand may be a full agonist for one cellular pathway and a full antagonist for another. This effect may result from ligand bias, in which ligands promote or bind to different conformational states of the receptor which then activates different cellular pathways. This situation applies to the β3-AR as it couples with different G-proteins that activate different pathways in different cell types [40,41]. SR59230A, originally used as a β3-AR antagonist based on its effects on cAMP accumulation, is now known to also have agonist properties, as shown by its ability to increase the extracellular acidification rate—a marker of activation for various intracellular pathways. β3-AR activation is associated with stimulation of the p38 mitogen-activated protein kinase (p38 MAPK) pathway and SR59230A mediates this process [42]. Intriguingly, p38 MAPK is linked to lipolysis in human adipocytes—a process mediated by β3-AR and inhibited by SR59230A [43]. What is more, SR59230A also triggers phosphorylation of extracellular signal-regulated kinases 1/2 (ERK1/2), and its efficacy in this pathway is even greater than that reported for the full agonist CL316243 [38]. Another study showed that in CHO K1 cells, β3-AR is coupled to both G(s) and G(i) proteins, leading to the activation of p38 MAPK and ERK in response to different ligands: zinterol (agonist), L755507 (partial agonist), and L748337 (antagonist) [44] (Table 1). Nonetheless, this issue represents an important aspect of β3-AR pharmacology. To the best of our knowledge, no clinical trials focusing exclusively on ligand-directed agonism of β3-AR agonists have been conducted yet.

### 2.4. β3-AR Agonists

#### 2.4.1. Structure–Activity Relationship of β3-AR Agonists

The structure–activity relationship is of the utmost importance to medicines [45]. While it can be unknown for some drugs, there are numerous examples in medicinal chemistry of an established association between molecule structure and its therapeutic or toxic effects [45,46]. What is more, nowadays, the structure–activity relationship is gaining more and more attention as an essential element of the drug discovery process [47]. In the case of β3-AR ligands, the chemical structure is very important: most of them share an ethanolamine scaffold (-OH–C_2_H_5_–NH–), which is recognized as the minimal pharmacophore for activity [48].

#### 2.4.2. Mirabegron and Vibegron—Agonists for Overactive Bladder Treatment

##### Mirabegron

Mirabegron (Figure 1a) is an EMA (European Medicines Agency)-approved medication used in the treatment of overactive bladder as monotherapy or in combination with an antimuscarinic agent [49,50,51]. It relaxes the detrusor muscle via β3-AR activation and subsequent cAMP production [52]. Mirabegron is selective for human β3-AR (Table 1) [53]. This selectivity was recently emphasized by Li et al. [54], who discovered the first turn-on fluorescent ligand (consisting of mirabegron conjugated to a fluorophore) that was highly selective for human β3-ARs. Mirabegron’s chemical structure includes a phenylethanolamine backbone that engages the orthosteric binding pocket. In addition, the 2-amino-thiazole tail may influence interactions with regions outside the canonical binding site, although the exact nature of these interactions remains under investigation (Figure 1a) [55].

Mirabegron is a molecule of low systemic toxicity and is well tolerated even in doses six-fold higher than therapeutic ones [56]. It shows significantly fewer adverse effects than solifenacin, a previous drug more commonly prescribed for the treatment of overactive bladder [57]. Most of its adverse effects are cardiovascular and resemble those produced by stimulation of β1 and β2-ARs [58,59]. At supratherapeutic doses, mirabegron may also activate β1-ARs in humans, as β1-AR blockade mitigates these effects [60]. Reported cardiovascular adverse effects include atrial fibrillation, palpitations, or increased heart rate and blood pressure leading to hypertension; however, during clinical trials, their incidence did not significantly differ compared to placebo groups [56,61]. Nonetheless, in 2015, the Medicines and Healthcare Products Regulatory Agency (MHRA) of the UK reported cases of severe hypertension including hypertensive crisis and cerebrovascular complications associated with mirabegron use. Since then, mirabegron is contraindicated in patients with severe, uncontrolled hypertension [62]. Furthermore, Mo et al. presented that mirabegron enhances norepinephrine release from sympathetic nerves resulting in a positive inotropic effect [63], and this mechanism is proposed to mediate cardiovascular adverse effects [64]. Mirabegron treatment does not significantly alter the corrected QT interval (QTc) in dogs, suggesting low arrhythmogenic potential [59] And in clinical trials, a dose of 200mg (which is four times higher than the therapeutic dose) slightly prolonged QTc in female participants only [65]. Apart from that, mirabegron may also act as an antagonist of α1-AR at high concentrations [66], and because of this mechanism, it relaxes human adipose tissue arteries [67]. Moreover, its vasodilative action on aortas isolated from transgenic mice expressing human β3-AR is linked to the β3-AR/G(i)/NOS pathway [68]. Nevertheless, these effects are clinically insignificant as mirabegron tends to increase blood pressure in clinical practice [69].

##### Vibegron

Vibegron (Figure 1b) is another β3-AR agonist approved by the EMA to treat overactive bladder [70]. It is considered superior to mirabegron because of its higher selectivity and its lack of interactions with cytochrome p450 enzymes [71]. Reports show that its efficacy at the β3-AR may approach 100% of that of isoproterenol [72] and it affects fewer β1- and β2-ARs compared to mirabegron [24,72,73]. In clinical trials, vibegron showed a similar incidence of adverse effects compared to placebo, with the highest incidence of hypertension [74,75,76]. Weber et al. presented that vibegron did not significantly change blood pressure assessed by ambulatory blood pressure monitoring in patients with overactive bladder when compared to placebo. Although it is worth noting that those results were given in 90%, not 95% confidence intervals [77]. Vibegron is an example of a medicine that lost the classical phenylethanolamine core used in earlier β-AR agonists in favour of a pyrrolidine core (Figure 1b) [78]. Aforementioned change was made to decrease toxicity as well as to reduce the risk of phospholipidosis [73].

#### 2.4.3. Nebivolol, Celiprolol, and Carvedilol—Third-Class β-Blockers and Their Influence on β3-AR

Nebivolol (Figure 1c), carvedilol (Figure 1d), and celiprolol (Figure 1e) are classified as third-generation β-blockers. They have additional properties that promote vasodilation, which distinguishes them from older, first- and second-generation β-blockers. Their vasorelaxant mechanism is multifaceted and most likely depends on vessel localisation and type [79,80]. It involves an increase in NO production, which, in the case of nebivolol and (to a lesser extent) celiprolol, occurs via β3-AR stimulation [81,82,83]. Additionally, nebivolol and carvedilol relax glomerular vessels by potentiating adenosine triphosphate (ATP) release in response to mechanical stimuli. ATP then binds to the P2Y receptor and increases NO production, leading to vasodilation [79].

##### Nebivolol

Nebivolol is a cardioselective β-blocker approved to treat hypertension [84] and also recommended by the European Society of Cardiology (ESC) to treat heart failure [85]. Key parts of the nebivolol scaffold include the ethanolamine segment linking to the bicyclic ring and 2H-chromen part (Figure 1c) [86]. The drug is available as a mixture of two enantiomers, L- and D-nebivolol, of which D-nebivolol is a selective β1-AR blocker [87] with additional features of an α1-AR antagonist and β2-AR agonist. Both enantiomers are β3-AR agonists, stimulating NO production [88]. Nebivolol also stimulates endothelial membrane-associated estrogen receptors, generating NO via similar cellular pathways to β3-AR responses [89], but this mechanism is secondary to the one mediated by β3-AR [90]. Moreover, metabolites of nebivolol can further stimulate NO production in different manners [80,91]. However, β3-AR stimulation seems to be the most relevant vasodilating mechanism, as the vasorelaxation is fully abolished in the receptor-deficient mice [82]. This mechanism was also observed for humans [92]. Notably, nebivolol also possesses antioxidative properties, alleviating oxidative stress and endothelial dysfunction, which is beneficial in many cardiovascular pathologies [93,94].

Baker indicated that nebivolol is a selective β1-AR antagonist that also performs a clinically insignificant partial agonist response towards the receptor at maximal efficacy of 2.8 ± 0.3% of isoprenaline. Moreover, it presents no cAMP response upon binding to β3-AR [21] (Table 1), which is in line with the study by Frazier et al. [95], who reported that nebivolol does not induce cAMP accumulation in CHO K1 cells transfected with human β3-AR. Additionally, nebivolol relaxes the human detrusor muscle independently of β3-AR stimulation. Different observations indicate that nebivolol induces both cAMP production in human umbilical vein endothelial cells (HUVECs) [96] and lipolysis via the p38 MAPK pathway in human adipocytes [43]. These notable inconsistencies may suggest that nebivolol acts as a “biased ligand” with very low potency for activating adenylyl cyclase and increasing cAMP levels. Although its action on β3-AR is well established, definitive evidence for ligand-directed agonism remains lacking.

##### Carvedilol

Carvedilol is indicated for the treatment of hypertension, coronary artery disease (CAD), and heart failure [97]. It is a non-selective β-blocker with similar binding properties to all β-ARs, including β3-ARs, which is its distinctive feature among β-blockers [98]. However, it also exerts vasodilatory effects resulting from α1-AR antagonism, confirmed in humans [99]. At higher concentrations, it is also a calcium channel antagonist; nonetheless, the antihypertensive significance of this mechanism is doubtful [100,101]. Carvedilol also alleviates endothelial dysfunction [102], as well as oxidative stress [103], because of its multifaceted antioxidative properties, which include direct scavenging of oxygen-free radicals, as well as inhibition of their release from activated neutrophils, inhibition of lipid peroxidation in cell membranes, and sustainment of endogenous antioxidants such as vitamin E and glutathione in an oxidative stress environment [104]. Structurally, carvedilol is a carbazole derivative containing a secondary alcohol and a secondary amine (Figure 1d). It has one chiral centre. The S(−) enantiomer is both a β- and α1-AR antagonist, whereas the R(+) enantiomer predominantly blocks α1-AR [105].

##### Celiprolol

Celiprolol is a cardioselective β-blocker with intrinsic sympathomimetic activity indicated for the treatment of hypertension and CAD [106]. It also increases NO production [107], but that mechanism of action involves partial agonism of β2-AR and possibly serotonin receptor 5-HT1a [108,109]. The drug also possesses modest inhibitory abilities towards α2-AR, which appears to be insignificant for its overall pharmacodynamics [110]. Moreover, celiprolol was shown to induce vasorelaxation of porcine coronary arteries by stimulating β3-AR [83]; nonetheless, the clinical significance of this finding has not been elucidated yet. Celiprolol is an aromatic ketone and has a 3-(tert-butylamino)-2-hydroxypropoxy side chain. The diethylurea moiety distinguishes this drug from classic ethanolamine β-AR antagonists (Figure 1e) [111].

## 3. Role of β3-AR in Blood Vessels

### 3.1. Endothelial Dysfunction

Endothelial dysfunction is a proinflammatory and prooxidative state characterized by impaired endothelial cell function resulting from endothelial activation [112]. Endothelial cells change their protein expression to promote adhesion of inflammatory cells, which subsequently leads to their activation and exacerbates endothelial dysfunction, as well as the progression of atherosclerosis. The endothelium itself can generate reactive oxygen species (ROS), for example, through endothelial NOS isoenzyme (eNOS) uncoupling [112]. In its uncoupled monomeric form, eNOS produces superoxide instead of NO, promoting oxidative stress, endothelial dysfunction, and ischemia–reperfusion injury. Another enzyme associated with oxidative stress is the reduced nicotinamide adenine dinucleotide phosphate (NADPH) oxidase (NOX), which produces superoxide, the main ROS. NOX is activated by angiotensin II, and once activated, it mediates glutathionylation of the β1 subunit of Na^+^/K^+^-ATPase, leading to impaired pump work. It also mediates glutathionylation of eNOS, resulting in its uncoupling [113]. By this mechanism, angiotensin II contributes to endothelial dysfunction and oxidative stress [114]. Eventually, the endothelium loses its vasorelaxant properties, which can result in arterial hypertension [112]. Treatment with the β3-AR agonist (CL3162430) restores normal function of Na^+^/K^+^-ATPase and eNOS and decreases glutathionylation by inhibiting NOX in male New Zealand white rabbits (Figure 2). Moreover, these proteins are more glutathionylated in β3-AR-deficient mice, which further suggests the key β3-AR role in proper endothelial function [115].

Nebivolol activates eNOS in the endothelium, increasing NO production [91,96]. Moreover, it prevents oxidative stress by suppressing NOX in both neutrophils and the endothelium, thus inhibiting NOS uncoupling (Figure 2) [93]. By diminishing ROS production, nebivolol indirectly raises NO bioavailability, further increasing its levels [116]. Furthermore, in rat aorta, nebivolol antagonizes the contractile effect of asymmetric dimethylarginine (ADMA) [117], which is an endogenous NOS inhibitor. It is elevated in many cardiovascular pathological conditions, exacerbating oxidative stress and eventually contributing to endothelial dysfunction [118]. In obese rats, nebivolol normalizes endothelial dysfunction by upregulating eNOS via AMPK and decreasing oxidative stress by upregulating catalase and superoxide dismutase (Figure 2). Additionally, it inhibits inflammation by inhibiting inflammasome activity, thus lowering interleukin-1β (IL-1β) levels (Figure 2) and, as a result, decreasing macrophage infiltration in the aorta and its remodelling, both in Sprague–Dawley rats [119] and New Zealand white rabbits [120]. Aortic remodelling occurs as a thickening of the tunica media due to smooth muscle proliferation. Nebivolol induces NO production, which inhibits ornithine decarboxylase, leading to a decrease in the production of polyamines (putrescine, spermine, and spermidine), which are important DNA protectors during the cell cycle. This inhibition stops the cell cycle and cell proliferation [121]. Antioxidative properties of nebivolol were confirmed in a human trial on hypertensive participants, where treatment reduced blood levels of oxidative stress markers, such as oxidized low-density lipoprotein (ox-LDL) or low-density lipoprotein (LDL) hydroperoxides [122].

In lipopolysaccharide (LPS)-stimulated macrophages, β3-AR agonisation with ICI215001 inhibits NADPH oxidase 2 (NOX2) activity and activates ERK, thereby increasing the expression of peroxisome proliferator-activated receptor γ (PPARγ). It induces expression of catalase (an antioxidative enzyme that degrades hydrogen peroxide) and inhibits NOX2, thus inhibiting the ROS-induced nuclear factor κ-light-chain-enhancer of activated B cells (NFκB). This further lowers NOX2 expression, as well as IL-1β and cyclooxygenase-2 (COX2), thus inhibiting the cell’s inflammatory response (Figure 2) [123]. Interestingly, β2-AR activation in LPS-stimulated macrophages also exerts anti-inflammatory effects [124]; however, in the absence of inflammatory triggers, its activation promotes inflammation [125]. Therefore, one can hypothesize that upon acute stress, catecholamines activate β2-AR in macrophages, promoting an inflammatory response. However, under chronic stress, β2-ARs are downregulated due to overstimulation, and β3-AR activation then suppresses the inflammatory response [123]. To the best of our knowledge, the effects of β3-AR stimulation in inactivated macrophages have not been assessed yet. The anti-inflammatory and antioxidative properties of β3-AR are in line with those of nebivolol. Furthermore, both β3-AR expression and nebivolol response are unchanged during inflammation [126]. β3-AR stimulation, therefore, might be useful and beneficial in cardiovascular therapy as many of these pathologies co-exist with inflammation and oxidative stress.

### 3.2. Atherosclerosis

High-density lipoprotein (HDL) is a major factor responsible for “reverse cholesterol transport”, a process in which it binds cholesterol particles from peripheral cells (such as macrophages/foam cells and epithelial cells) via export mediated by ATP-binding cassette A1 (ABCA1) or G1 (ABCG1) and transports it to hepatocytes where it enters the cell via scavenger receptor B1 (SR-B1) (Figure 2) [127]. This concept was thought to protect against atherosclerosis; however, the pharmacological increase in HDL levels did not meet clinical significance. Novel concepts hypothesize that increasing levels of apolipoprotein AI (ApoAI), which is a key component of HDL (Figure 2), might be beneficial for atherosclerotic patients [128].

BRL37344 (Table 1) in apolipoprotein E (ApoE)-deficient mice (animal model for hypercholesterolaemia and atherosclerosis) significantly reduces serum total cholesterol levels and deposition of free cholesterol, as well as cholesterol esters in the rat aorta (indicator of aortic atherosclerosis) [129], along with serum triglycerides, total cholesterol, and the VLDL/LDL ratio. It also increases serum HDL levels and the HDL/total cholesterol ratio [130]. It has been shown that β3-AR stimulation with BRL37344 increases the expression of ApoAI and decreases apolipoprotein AII (ApoAII) in hepatocytes [131]. It involves the cAMP/PKA pathway which enhances the activity of the ApoAI gene promoter by activating hepatocyte nuclear factors 3 and 4 (HNF-3/4) and PPARγ (Figure 2) [132]. Some researchers also showed activation of peroxisome proliferator-activated receptor α (PPARα) [129]; however, others presented that PPARα expression is inhibited by β3-AR [132]. While β3-AR agonism has no proven effect on the cholesterol content of foam cells, the upregulated ApoAI stimulates the expression of ABCA1, thus activating cholesterol efflux, decreasing its content, and afterwards increasing serum HDL levels [131] (Figure 2). Eventually, in ApoE-deficient mice, BRL 37344 decreases aortic atherosclerotic lesion areas and attenuates fibrous cap formation, suppressing lesion progression, thus alleviating atherosclerosis. Nevertheless, the effect is weaker than the one of atorvastatin [130]. However, the preclinical outcomes of mirabegron treatment are inconsistent with the findings described above. In high-fructose and high-fat diet rabbits, despite antioxidative effects, ameliorating insulin sensitivity (without affecting fasting glycemia), heart contractile properties, and the vasorelaxant response to insulin of carotid arteries, mirabegron showed no effect on the lipid profile, as well as significantly worsened the area and stage of aortic atheroma [133].

Zinc-α-2-glycoprotein is an adipokine and β3-AR agonist [134] with a not yet fully understood function. Its levels are diminished amongst patients with CAD compared to healthy controls and negatively correlate with BMI, fasting blood glucose, lipid profile (triglycerides and LDL), and Gensini score (severity of CAD). Moreover, lower levels of Zinc-α-2-glycoprotein are independent predictors of CAD. Zinc-α-2-glycoprotein colocalizes with β3-AR and macrophages in the centre of atherosclerotic plaques, resulting in lower serum levels. Furthermore, through β3-AR, it inhibits cytokine release from LPS-stimulated macrophages and promotes their M2 phenotype, which has more anti-inflammatory properties. The receptor activated by the zinc-α-2-glycoprotein on macrophages inhibits phosphorylation of c-Jun N-terminal kinase (JNK) and activator protein 1 (AP1) expression without affecting NFκB, p38 MAP, or ERK1/2 [135].

Despite the above, β3-AR stimulation might also have deleterious effects in atherosclerosis. The atherosclerotic model of ApoE-deficient mice revealed accelerated progression of atherosclerosis after myocardial infarction. The plaques had larger lipid necrotic cores with thinner fibrous caps, an increase in activated M1 macrophages, and higher activity of proteases in the plaque. All of this contributes to a more unstable plaque phenotype, increasing the likelihood of rupture. Splenic monocytopoesis was also increased. The β3-AR antagonist (SR 59230A) and sympathetic blockade with 6-hydroxydopamine mitigated all those changes. Authors conclude that after myocardial infarction, sympathetic activation promotes inflammation partially via β3-AR on monocytes and their progenitors, which leads to the rapid progression of atherosclerosis and higher risk of another cardiovascular event [136]. Tjandra et al. showed that by the same mechanism, β3-AR contributes to systemic bone weakening and a loss of structure after myocardial infarction. Inhibition of the receptor moderately reduced bone demineralisation, suggesting additional involvement of other mechanisms [137].

Bubb et al. presented that β3-AR stimulation with CL 316243 mediated beneficial antioxidative effects and improved blood flow in ischaemic limbs of type 1 and 2 diabetic mice models [113]. Because of promising preclinical outcomes, authors proceeded with a clinical trial named STAR-PAD (stimulating β3-AR for peripheral artery disease) with mirabegron as intervention (50 mg/day orally) in patients suffering from intermittent claudication [138].

### 3.3. β3 Adrenergic Receptor and Blood Pressure

#### 3.3.1. Pulmonary Hypertension

Pulmonary hypertension (PH) is clinically defined as a condition in which mean pulmonary arterial pressure exceeds 20 mm Hg, and pulmonary vascular resistance (PVR) is greater than 2 Wood units [139]. Impairment of cyclic guanosine monophosphate (cGMP) signalling is a key feature of PH. Within the pulmonary vasculature, loss of proper cyclic nucleotide function may cause many adverse effects, such as obstructed vasodilation, poorly controlled smooth muscle cell proliferation, and platelet aggregation, thereby contributing to PH [140]. Despite advancements in understanding this condition, the development of specific pharmacological treatments remains an active area of research. Recent investigations have focused on β3 adrenergic agonists to evaluate their potential to improve haemodynamics in PH as β3-AR increases cGMP levels via NO. A study by García-Álvarez et al. evaluated the effects of the β3-adrenergic agonist, mirabegron, in patients with PH secondary to heart failure (HF), compared to placebo controls [141,142]. In this double-blind, randomized clinical trial, the results showed no significant reduction in PVR. Additionally, no improvements were observed in patients’ clinical status or their subjective assessment of quality of life. However, a notable improvement in right ventricular (RV) ejection fraction, measured by cardiac magnetic resonance, was observed in patients treated with mirabegron compared to placebo. This was a secondary outcome of the study, suggesting a potential benefit, though insufficient as standalone evidence of mirabegron’s efficacy [140]. On the other hand, a study conducted by Sun J et al. explored the effects of the β3-AR antagonist, SR59230A, in pulmonary arterial hypertension (PAH) and subsequent HF in a rat model of PAH induced by monocrotaline [143]. SR59230A treatment significantly improved RV function and reduced lung β3-AR expression, which were markedly elevated in the lung and heart tissues of PAH rats. Notably, SR59230A alleviated lung inflammation and decreased elevated expression of eNOS without altering inducible NOS isoenzyme (iNOS) levels [143]. Therefore, β3-AR blockade with SR59230A may help reduce lung structural remodelling and inflammation in PAH by lowering oxidative stress and improving heart function.

A randomized, double-blind trial by Bundgaard et al. investigated the haemodynamic effects of mirabegron in patients with severe HF with reduced ejection fraction (HFrEF) [144]. After one week, treatment with 300 mg of mirabegron significantly increased the cardiac index and decreased PVR compared with placebo. Other haemodynamic parameters, including stroke volume and blood pressure, showed no significant differences. While mirabegron showed some potential in improving pulmonary haemodynamics, further research is needed to confirm its clinical efficacy for PH. Additional insights came from an experimental study by Ana García-Álvarez et al., which explored the effects of β3-adrenergic agonists in chronic PH in pigs [11]. BRL37344 (Table 1) induced a significant acute reduction in PVR shortly after administration, demonstrating a promising haemodynamic effect. Furthermore, after two weeks of treatment, both BRL37344 and mirabegron significantly reduced PVR [11]. Magnetic resonance imaging also showed improved RV performance [145]. Moreover, histological analysis performed in this investigation revealed decreased levels of Ki67, a cellular proliferation marker, and increased expression of p27, a cyclin-dependent kinase inhibitor that halts cell cycle progression from G1 to S-phase. These findings suggest reduced vascular proliferation in pulmonary arteries, potentially mitigating perfusion obstruction. These results align with previous research on the role of p27 in PH [11], which is a key cyclin-dependent kinase inhibitor that regulates anti-proliferative effects on various vascular cells [146].

#### 3.3.2. Arterial Hypertension

Arterial hypertension is defined as persistently elevated blood pressure of 140/90 mmHg or higher [147]. This condition primarily arises from the dysfunction in cardiovascular regulatory systems, leading to increased cardiac output and/or elevated systemic vascular resistance. Contributing factors include enhanced vasoconstriction, reduced vascular elasticity, and impaired sympathetic nervous system function [147,148]. Studies have demonstrated that β3-ARs play a crucial role in lipolysis, thermogenesis, and smooth muscle relaxation [149]. Specific β3-AR agonists, such as CL 316243 and amibegron, reduce cardiac contractility and decrease both afterload and preload in animal models [150,151]. The vasodilatory effect of β3-ARs was demonstrated by isoprenaline-induced relaxation of phenylephrine-preconstricted rat aortic rings, which was further enhanced by β3-agonists amibegron and CGP 12177, and prolonged by nadolol [152]. Moreover, Berg et al. studied the roles of β-AR subtypes in hypertension using selective antagonists in normotensive and hypertensive rats [153]. In normotensive rats, blocking peripheral β1/3-AR increased α1-AR-mediated vasoconstriction. In hypertensive rats, β1/3-AR inhibition enhanced nerve-activated vasoconstriction, indicating a complex role of these receptors in blood pressure regulation.

Emerging research highlights the therapeutic potential of β3-AR in promoting relaxation of the vascular smooth muscle in blood vessels [7]. A study conducted by Saunders SL et al. proved the presence of β3-AR subtypes in the endothelium of the cremaster muscle artery of rats [154]. The study also showed that mirabegron and CL 316243 induced β3-AR-mediated vasodilation only when β1/β2-ARs were blocked (using nadolol, atenolol, and ICI-118551), whereas amibergon and isoprenaline-induced dilation was inhibited under the same conditions. This suggests that constitutive β1/β2-AR activity can suppress some β3-AR function (mediated by mirabegron or CL31243) in the endothelium of skeletal muscle resistance arteries [154]. Vasodilation in this setting depends not only on NO and cGMP production [152,154] but also on the activity of calcium-activated potassium channels, whose blockade fully abolished the effect. Additionally, β3-AR inhibition appears to be mediated by PKA, as PKA inhibition restores the vasodilatory response [154]. Moreover, a previous study showed that amibegron significantly reduced mean aortic pressure in dogs, with a more pronounced effect in hypertensive animals [151]. In another study, Sheng LJ et al. revealed that β3-AR in perivascular adipose tissue plays an important role in the vascular progression of hypertension [155]. In a deoxycorticosterone acetate (DOCA)-salt-induced hypertensive mouse model, inhibition of the β3 receptor with SR59230A visibly reduced the expression of uncoupling protein 1 and β3-AR in aortic perivascular adipose tissue. SR59230A treatment caused a significant reduction in lipid droplets in the tissue, which could disrupt its regulation of vascular tone and vessel remodelling. Moreover, collagen staining revealed vascular dissection in the aortic wall after β3-AR inhibition with SR59230A [155]. This suggests that inhibition of the β3-AR in DOCA-salt-induced hypertensive mice may promote vascular injury, leading to arterial dissection.

Another outcome was observed in a study conducted by Ling S et al. on high-salt-treated rats, where β1 and β3-AR pathways influenced the expression of certain miRNAs [156]. Microarray analysis revealed increased miR-320 expression as well as decreased miR-26b and miR-21 levels in the rats’ aorta. These changes were restored to normal levels after the administration of nebivolol, with similar effects observed using atenolol combined with BRL37344. Insulin Growth Factor-1 Receptor (IGF1R) was identified as a target of miR-320, and Phosphatase and Tensin Homolog on chromosome ten (PTEN) was identified as a target of miR-26b and miR-21. Nebivolol treatment and direct antisense inhibition of miR-320 increased IGF1R expression, while miR-320 overexpression reversed this effect. Simultaneously, PTEN levels decreased after nebivolol administration or miR-26b and miR-21 overexpression, whereas inhibition of miR-26b or miR-21 attenuated the effect of nebivolol. These findings suggest that nebivolol may regulate disrupted miRNAs in hypertension by restoring aberrant expression of IGF1R and PTEN. Their regulation could improve impaired vascular Akt (protein kinase B)/eNOS signalling [156], which is known to influence arterial hypertension [152], making β3-ARs potential therapeutic targets.

#### 3.3.3. Preeclampsia

Preeclampsia is defined by the American College of Obstetricians and Gynaecologists (ACOG) as blood pressure greater than or equal to 140/90 mmHg accompanied by proteinuria or other signs of maternal organ dysfunction after 20 weeks of gestation. Its pathophysiology is associated with low NO bioavailability [157]. Bueno-Pereira et al. investigated the effects of nebivolol on NO synthesis in endothelial cells exposed to plasma from preeclampsia patients [158]. Endothelial cells were incubated with plasma from preeclampsia patients and exhibited significantly lower NO levels compared to those incubated with plasma from healthy pregnant women. Interestingly, after nebivolol treatment, the cells revealed increased NO levels via the β3-AR/eNOS pathway [158]; therefore, theoretically, nebivolol might be a beneficial treatment for preeclampsia.

#### 3.3.4. Portal Hypertension

Portal hypertension arises from distorted hepatic haemodynamics, such as increased intrahepatic vascular resistance and portal blood flow, and is therefore commonly associated with liver cirrhosis [159]. The hepatic venous pressure gradient measures the difference between the portal and hepatic vein pressure [160]. When this gradient exceeds 10 mmHg, it indicates portal hypertension [161]. This increases the risk of developing complications associated with liver cirrhosis, such as oesophageal variceal bleeding. Therefore, effective treatments to mitigate these consequences could be highly beneficial. Immunohistochemical analysis confirmed the presence of β3-AR in the liver and portal vein of both cirrhotic and healthy rats [161]. Intravenous administration of amibegron in cirrhotic rats dose-dependently reduces portal pressure (PP), with the highest dose lowering PP to approximately half of its baseline value, without a significant effect in the healthy animals. The action is completely inhibited by β3-AR blockage with SR59230A, providing additional evidence for the mechanism. In all animals, central venous pressure remains unaffected during treatment. Furthermore, amibegron induces a concentration-dependent relaxation of the isolated portal vein, more pronounced in cirrhotic rats and also entirely blocked by SR59230A [161].

β3-AR-mediated vasodilation may occur independently of NO via activation of the G(s) protein-cAMP pathway or the inhibition of the RhoA/Rho-kinase pathway in smooth muscle cells [162]. β3-AR stimulation with CGP12177A reduces hepatic Rho-kinase activity and increases cAMP levels in cirrhotic rats [163], suggesting a vasodilation mechanism independent of NO. At the same time, β3-AR expression was significantly elevated. CGP12177A lowers PP without affecting systemic haemodynamics, while the antagonist, SR59230A, increases systemic vascular resistance and reduces cardiac output [163]. These findings highlight β3-AR agonists as promising candidates for the treatment of portal hypertension.

## 4. Role of β3-AR in the Heart

The β3-AR is present, amongst others β-ARs, in both heart atria and ventricles. Their stimulation exerts haemodynamic as well as metabolic effects [8,68,164]. Despite intensive studies, the exact mechanism and significance of many of these effects still lack understanding, especially since β3-AR responses in the heart may differ in physiological and pathological states [165,166]. In this section, we discuss the mechanism and potential clinical significance of β3-AR-mediated heart effects in both physiological and pathological environments.

### 4.1. Stimulation Effects and Ion Channels

Gaunthier et al. demonstrated the presence of functional β3-ARs in human heart ventricles. In their study on endomyocardial biopsies, β3-AR agonists administered together with nadolol induced a concentration-dependent negative inotropic effect. They also reduced both action potential duration (APD) and amplitude, although with different potencies (BRL37344 > amibegron = CL316243 > CGP12177) [8]. Later, it was shown that the receptor is coupled with several ion channels that contribute to the generation of the cardiomyocyte action potential.

#### 4.1.1. Hyperpolarization-Activated Current (I_f_)

The I_f_ current, also referred to as the “funny current,” is a pacemaker current in the heart. It is mediated by the opening of hyperpolarization-activated, cyclic nucleotide-gated (HCN) channels during hyperpolarization, leading to increased membrane permeability for sodium and potassium ions, resulting in slow depolarization. Physiologically, its activation is facilitated by β1/2-AR stimulation, contributing to positive chronotropism [167]. In rat cardiomyocytes, however, β3-AR stimulation with amibegron, acting via G(i) proteins and NO, reduced I_f_ current amplitude by shifting its activation to more negative potentials and lowering its maximum conductance. This effect was more pronounced in cells after ischemia and barely detectable in normal cardiomyocytes [166].

#### 4.1.2. Inward Rectifier Potassium Current (I_K1_)

The resting membrane potential of cardiomyocytes is primarily maintained by the inward rectifier potassium current (I_K1_), which also participates in the late repolarization phase. I_K1_ activation increases resting membrane potential and shortens APD [168]. It is generated by potassium efflux through channels from inwardly rectifying the potassium channel (Kir) family present in cardiomyocytes (Kir2.1–2.3) [169], vascular smooth muscle cells (Kir2.1), and endothelial cells (Kir2.1–2.2) [170]. Scherrer et al. [171] showed that in the Xenopus oocyte transfected with genes encoding these channels together with the human β3-AR gene, stimulation of the receptor with isoproterenol increased the current mediated by Kir2.1 and Kir2.2 through activation of protein kinase C (PKC) and protein kinase A (PKA), respectively. The effect was also significant for heterodimer channels consisting of Kir2.2 and other subtypes, which are thought to produce the I_K1_ current in cardiomyocytes [172]. Those current alterations may participate in β3-AR-induced APD reduction. Nevertheless, in rabbit atrial cardiomyocytes, I_K1_ decreased after β3-AR stimulation with BRL37344. In contrast, the opposite effect was observed in cardiomyocytes from animals with atrial fibrillation [165].

#### 4.1.3. Transient Outward Potassium Current (I_to_)

The early repolarization phase is mostly mediated by fast potassium efflux, known as transient outward potassium current I_to_, through voltage-gated potassium channels (Kv) expressed in cardiomyocytes and activated by voltage depolarization [173]. Its activation shortens APD and has been observed after β3-AR stimulation with BRL37344 in cardiomyocytes from the atrial fibrillation (AF) rabbit model, but not in healthy controls [165].

#### 4.1.4. Slow Delayed Rectifier Potassium Current (I_Ks_)

KvLQT1/minK is a voltage-gated ion channel that mediates a slow delayed rectifier potassium current (I_Ks_) responsible for the maintenance of the plateau phase of cardiomyocyte action potential. Its inhibition prolongs the plateau phase and APD, while its activation has the opposite effect [168]. In Xenopus oocytes expressing human β3-AR and KvLQT1/mink channels, isoproterenol agonized β3-AR and increased the I_Ks_ current via the G(s) protein. A similar effect was observed in guinea pig cardiomyocytes; however, the authors did not specify which part of the heart the cells originated from [174]. Nevertheless, these results are negated by another study on guinea pigs’ ventricular cardiomyocytes, where BRL37344 or isoproterenol, after β1/2-AR blockage, inhibited the I_Ks_ current, thus prolonging APD [175].

#### 4.1.5. L-Type Calcium Current (I_CaL_)

The L-type calcium current (I_CaL_) channels (known as dihydropyridine channels) could be found in cardiomyocytes, neurons, endocrine cells, and other cells all over the body [176]. They play a pivotal role in the mediation of calcium influx and the regulation of cardiac action potential [177]. In ventricular and atrial cardiomyocytes, they are activated by membrane depolarization and are responsible for maintaining the plateau phase [178]. In healthy cardiomyocytes, β3-AR activation increased I_CaL_, whereas the opposite effect was observed in experimental groups with induced AF [165]. Nonetheless, there are some inconsistencies regarding whether β3-AR truly modulates I_CaL_ channels. Interestingly, activation of the current has been reported at room temperature but not at physiological temperature [179,180]. Authors therefore suggest that, physiologically, the current remains unchanged under β3-AR stimulation, and the often-reported inotropic positive responses to BRL37344 are mediated by β1 and β2-ARs [180]. Those findings may explain the distorted results of some preclinical studies that investigated the role of β3-AR activation on I_CaL_ channels. Although I_CaL_ channels are implicated in numerous cardiac pathologies, such as heart failure, cardiac hypertrophy, and cardiac arrhythmias [177], the current inconsistencies make it difficult to draw definitive conclusions about their relationship to β_3_-AR signalling.

#### 4.1.6. Cystic Fibrosis Transmembrane Conductance Regulator (CFTR)

The cystic fibrosis transmembrane conductance regulator (CFTR) is a chloride channel expressed in cardiomyocytes. Its activation is independent of time and membrane potential but instead relies on β-AR stimulation and PKA-mediated phosphorylation [181]. Evidence suggests that β3-ARs are also functionally coupled to this chloride channel: activation of the receptor with BRL37344 increases the channel current and thereby shortens the APD, although through a mechanism that appears to be PKA-independent. This functional coupling is further influenced by receptor density, with high β3-AR expression leading to constitutive channel activation [182]. Interestingly, in endomyocardial biopsies from cystic fibrosis patients, BRL37344 did not shorten the APD, further supporting the relevance of this coupling [182].

#### 4.1.7. β3-AR in Atria

Interestingly, a positive chronotropic effect in isolated rat atria, traditionally associated with β1-AR activation [183], was also reported following β3-AR stimulation with both ZD7114 and ICI215001 [184]. Nonetheless, the effect coexisted with a cAMP increase, suggesting concomitant activation of other β-ARs, particularly since β1/2-ARs were not blocked. Moreover, similar effects were observed in dogs due to baroreflex activation [185]. Cohen et al. reported that in the presence of propranolol, Cl316243, which presents high affinity towards the rat β3-AR and other agonists (L739574 and its derivatives synthesized by authors), produced no positive chronotropic effect [27]. Nevertheless, in isolated rat hearts, BRL37344 is still able to induce positive chronotropism despite propranolol treatment, thereby excluding stimulation of the baroreflex mechanism and β1-AR stimulation [186]. These findings suggest that β3-ARs may contribute to positive chronotropism and that this effect depends on the specific agonist used.

### 4.2. Cardiac Arrhythmias

Cardiac arrhythmias occur when a localized region of the myocardium initiates action potentials independently of the sinoatrial node, which can result in disorganized cardiac contractions. They are associated with electrophysiological alterations in which β-ARs (including β3-AR) play a crucial role, as complete β-AR knockout has been shown to reduce arrhythmic susceptibility [187]. Interestingly, the contribution of β3-ARs to arrhythmogenesis may not be limited to their expression in cardiomyocytes, as the receptor is also present in adipose tissue. Epicardial adipose tissue (EAT), a complex endocrine and paracrine organ surrounding the myocardium [188], is a well-known participant in cardiac arrhythmogenesis [189]. In Babakr’s study [190], dissected human atrial trabeculae were cultured either with or without EAT and subsequently treated with BRL37344. The agonist increased the number of spontaneous contractions—mimicking arrhythmic events—only in trabeculae incubated with EAT, suggesting that β3-AR stimulation within EAT may promote cardiac arrhythmias [190].

#### 4.2.1. Ventricular Tachycardia

Ventricular tachycardia (VT) is a clinically common, potentially fatal, cardiac rhythm disturbance that originates from an abnormal electrical circuit in the ventricular myocardium. It is usually connected with channelopathies or structural abnormalities in the heart [191]. Considering that β3-AR activation may alter the function of ion channels and therefore electrical function of the heart, studies researching the role of β3-AR activation in VT were performed.

In both canine and rat VT models, β3-AR stimulation reduced the arrhythmia incidence [192,193]. It was mediated by the normalization of cellular calcium circulation, including a reduction in sarcoplasmic reticulum calcium load and leak [193]. Moreover, it downregulated the sodium–calcium exchanger (NCX) [192,193], which mediates calcium efflux from the cell during calcium overload states, such as in HF. The treatment also shortened the QTc interval; however, there was no effect on potassium channels or on β1/2-AR expression [192].

#### 4.2.2. Atrial Fibrillation

Atrial fibrillation (AF) is the most common cardiac arrhythmia. Its pathophysiology encloses deleterious atrial, electrophysiological, and histological remodelling [194], which is exacerbated by oxidative stress, leading to myocyte apoptosis and interstitial fibrosis [195]. Interestingly, in this condition, β3-AR is upregulated in human atria [196], and its influence on ion channels may differ from the physiological state [165], as mentioned above. Moreover, AF is a commonly reported adverse effect of mirabegron in clinical trials [74,75,76]; therefore, exploring the role of β3-AR in the pathophysiology of AF is of considerable interest.

##### Electrophysiological Alterations

One of the electrophysiological parameters altered during atrial fibrillation (AF) is the atrial effective refractory period (AERP), which represents the time during which a cardiomyocyte cannot generate a new action potential as it re-establishes normal ion concentrations across its cell membrane. This period shortens as the heart rate increases and returns to baseline as the rate decreases, producing corresponding changes in action potential duration (APD)—a phenomenon known as rate adaptation of refractoriness. In patients with AF, this adaptation is impaired: their AERP remains relatively short regardless of whether the arrhythmia is present. This loss of rate adaptation reflects irreversible atrial electrophysiological remodelling [197]. Due to its electrophysiological properties, β3-AR stimulation with BRL37344 shortens AERP and APD [165,198], as well as increases AF inducibility and arrhythmia duration, both in rabbits [164,165,198] and dogs [195] (Figure 3). In rabbits subjected to one week of atrial rapid pacing to induce AF, β3-AR stimulation further impaired the adaptation of refractoriness [165]. Interestingly, despite β3-AR upregulation in AF, its expression is even further increased after stimulation with the agonist, whereas inhibition of the receptor attenuates this upregulation [164,195,199]. Moreover, the β3-AR antagonists, SR59230A or L748337, mitigate all the above deleterious electrophysiological changes [195,198].

##### Structural Alterations

Progressive worsening of atrial haemodynamic parameters occurs in AF, primarily due to atrial dilation and loss of its ejection fraction. At the microscopic level, cardiomyocyte death and interstitial fibrosis appear, leading to further electrophysiological changes [194]. Those processes are exacerbated by β3-AR stimulation with BRL37344, as it has been shown on animal models to increase cardiomyocyte filament disintegration along with mitochondrial swelling [198], increasing apoptosis and interstitial fibrosis [195,198] (Figure 3). In a study on a dog model, atrial remodelling was partly driven by p38 MAPK pathway activation [199] and, in all cases, the β3-AR antagonist, L748337, diminished those remodelling changes, alleviating disease phenotype.

##### Biochemical Alterations

In AF, β3-AR activation aggravates oxidative stress through various mechanisms. In the canine model of AF, its stimulation with BRL37344 increased ROS production via upregulating iNOS [195] and downregulating key ROS-scavenging enzymes, including glutathione peroxidase [199], eNOS, and guanosine triphosphate cyclohydrolase-1 [195] (Figure 3). The latter one synthesizes tetrahydrobiopterin (BH4), an important NOS cofactor for NO production. The β3-AR antagonist, L748337, restores these enzymatic alterations [195]. Additionally, metabolic alterations have been observed, as demonstrated in the rabbit model, where atrial fibrillation leads to reduced levels of adenosine triphosphate (ATP) and total adenine nucleotides. These reductions are further exacerbated by BRL37344, indicating increased energy consumption and impaired energy production [164,198]. Indeed, in these cells, proteins involved in the transport of energetic substances such as fatty acid transporter CD36 (cluster of differentiation 36), carnitine palmitoyltransferase-1 (CPT1), and glucose transporter 4 (GLUT4) are downregulated (Figure 3), likely as a result of PPARα/PGC-1 pathway inhibition [164]. Moreover, impaired mitochondrial function, characterized, among other features, by ATP-synthase inhibition or downregulation of the respiratory chain elements, suppresses energy production [198] (Figure 3). SR59230A and L748337 restore normal metabolic function in those cardiomyocytes [164,198].

Altogether, these findings suggest that in the context of AF, β3-AR agonism may not be beneficial; in fact, it may even be detrimental, thereby supporting the rationale for β3-AR antagonism instead. These mechanistic insights may partially explain the outcomes of the post hoc analysis of the SENIORS trial, which reported that in elderly patients with heart failure and coexisting AF, nebivolol treatment did not reduce the incidence of the primary endpoints—namely all-cause mortality or cardiovascular hospitalization [200].

### 4.3. Myocardial Ischemia

β3-ARs are expressed in coronary micro-arteries [201], where their NO-mediated vasorelaxant effect enhances myocardial perfusion. Therefore, β3-AR agonism may be beneficial in CAD, a condition in which myocardial blood supply and coronary flow reserve are impaired (Figure 3). Nebivolol treatment increases NO production in coronary arteries via β3-AR [82]. In CAD patients, the drug increases coronary flow reserve by enhancing maximal coronary flow. It also reduces the collateral flow index, which means that the myocardium perfusion relies less on collateral coronary circulation and more on the main coronary arteries, likely due to their dilation [202]. Furthermore, β3-AR triggers angiogenesis via NO in the mice’s aortic rings [82] and in the culture of blood-derived endothelial colony-forming cells collected from CAD patients, but not from healthy individuals [113]. This might result in the better development of collateral circulation, potentially alleviating CAD symptoms (Figure 3). Nonetheless, in the study conducted by Hung et al. on CAD patients, nebivolol treatment led to greater luminal reduction and constructive remodelling of coronary arteries, with no difference in endothelial function when compared to the atenolol-treated group [203]. The authors concluded that, in the context of CAD, a 5 mg dose of nebivolol (approved for clinical use) exerts a minimal, if any, effect on coronary endothelial β3-AR stimulation.

In rat hearts, β3-AR expression exhibits circadian variation, with the highest levels observed at subjective noon and the lowest at subjective midnight, which correlates with the response to BRL37344. However, this circadian oscillation is abolished by myocardial infarction, resulting in a constant expression level throughout the day that resembles the noon levels observed in healthy controls, albeit with a reduced response to BRL37344 [186]. Nonetheless, β3-AR expression is generally increased in myocardial infarction. Its stimulation activates many intracellular signalling proteins, among which eNOS and neuronal NO synthase (nNOS) are the most important [145]. This leads to increased NO production, which is typically diminished during ischemia due to eNOS inhibition (Figure 3). Furthermore, β3-AR inhibits iNOS, whose activation is a marker of oxidative stress and the proapoptotic state [204]. Additionally, NOS inhibition completely mitigates the cardioprotection of β3-AR agonists. β3-AR might activate eNOS via Akt kinase, which also delays mitochondrial permeability transition pore (mPTP) opening—an important part of the mitochondrial apoptotic pathway [205]. Akt kinase also plays other anti-apoptotic and anti-inflammatory actions, as it decreases caspase 3 activity (an important enzyme in apoptosis), activates ERK, whose expression significantly correlates with infarct size reduction, and inactivates glycogen synthase kinase-3β (GSK3β), which is a proapoptotic protein enabling mPTP opening (Figure 3). Nevertheless, the latter two effects seem unnecessary for significant cardioprotection [206,207,208]. Additionally, stimulation of β3-AR after myocardial infarction might be beneficial as an antiplatelet therapy. In the study of Mendes-Silvério et al. [209], mirabegron inhibited platelet aggregation by increasing cAMP levels and decreasing levels of calcium cations, as well as thromboxane in the platelet cytoplasm. Although mirabegron concentrations used during the study were higher than those reached in clinical use, this might be an interesting direction for future research.

Heart ischemia–reperfusion injury (IRI) occurs when blood flow to the myocardium is restored after an episode of ischemia. The blood reflow is crucial in dealing with ischaemic conditions, such as myocardial infarction; however, reperfusion itself initially exacerbates damage contributing to a larger infarct size. Therefore, discovery of techniques to manage IRI is of the highest clinical interest. A key role in IRI is played by ischaemic-damaged mitochondria, which start to produce ROS in response to oxygen flux. This increases oxidative stress and leads to inflammation and cardiomyocyte death [210]. Therefore, therapies alleviating oxidative stress are sought. Animal studies present that cardiac but not epithelial β3-AR stimulation has a cardioprotective effect in IRI, as mirabegron and CL316243 reduce infarct size and ameliorate haemodynamic function [68,145] (Figure 3). Nebivolol with BRL37344, apart from these properties, additionally decreases fibrosis and preserves contractile function [10,207]. Interestingly, both receptor-deficient and overexpressing mice present smaller infarct sizes than the wild type [68,145]. The receptor overexpression remodels mitochondrial network in the cell, leading to better survival of cardiomyocytes during IRI (Figure 3). It promotes mitochondrial biogenesis, thus increasing mitochondrial number; however, these mitochondria are smaller and contain more uncoupling protein 2 (UCP2), which diminishes ATP production in the respiratory chain in favour of heat. Nevertheless, total ATP production in the left ventricle is not mitigated. This protein is also cardioprotective as it lowers ROS production during reperfusion and is downregulated after, thus mitochondria are coupled, resulting in their proper function and lower mitophagy. Eventually, cardiomyocytes are more resistant to cell death and express more anti-apoptotic and antioxidative proteins [68].

β3-AR appears to mediate the cardioprotective effects of exercise training, contributing to a reduction in infarct size. Notably, β3-AR expression increases in response to exercise and remains elevated for up to one week following cessation of training [211]. In mice, exercise-induced catecholamines stimulate β3-AR, which activates AMP-activated protein kinase (AMPK) [212]. This kinase then promotes binding of heat-shock protein 90 (HSP90) with eNOS, preventing its uncoupling and providing cardioprotection. This interaction is absent in β3-AR knockout mice, where exercise training induces eNOS uncoupling, thus exacerbating IRI. eNOS activation also increases the production of NO metabolites, nitrite, and nitrosothiols, which can provide NO during an ischaemic state and might be blood markers for cardioprotective potency. Another effect is the increased expression of superoxide dismutase, which further diminishes oxidative stress [211,212]. In obesity and type 2 diabetes mouse models, the β3-AR/eNOS/NO pathway is defective; therefore, BRL37344, as well as epinephrine, show no effect on both infarct size and eNOS expression. In these models, exercise training decreases the expression of all NOS isoforms, leading to reduced levels of NO metabolites. However, it alleviates oxidative stress while still reducing the infarct size. In non-training subjects, eNOS is uncoupled and iNOS activity is elevated, suggesting nitrosative stress [213].

β3-AR also plays a role in deleterious cardiac remodelling, which may lead to HF [214]. Exacerbated negative inotropic and lusitropic effects may occur after a myocardial infarction, leading to an increase in left ventricular end-diastolic pressure (LVEDP) and further worsening cardiac dysfunction. Therefore, β3-AR antagonism may represent an additional therapeutic option (Figure 4). In rats, after myocardial infarction without reperfusion, nebivolol treatment prevented those changes by inhibiting heart β3-AR upregulation and other β-AR alterations, which was confirmed by the preservation of the heart inotropic positive response to isoproterenol altered in non-treated groups. Authors suppose that early nebivolol treatment in patients after acute coronary syndrome might be beneficial for cardiac function [215]. Additionally, in cardiomyocytes, after infarction, the I_f_ current is upregulated, which may mark electrophysiological remodelling, leading to HF. As amibegron inhibits these currents in the rat model, β3-AR stimulation might attenuate this remodelling [166] (Figure 3). On the other hand, treatment with carvedilol downregulates β3-AR and G(i)-protein levels in diabetic rat hearts after myocardial infarction. This downregulation correlates with the reduction in pathological cardiac remodelling [216]; therefore, the authors suggest that it underlies the therapeutic effects of carvedilol, which were clinically confirmed by the CAPRICORN echocardiographic substudy [217].

### 4.4. Heart Failure

Heart failure (HF) is characterized by increased sympathetic activity [218]. In a failing myocardium, the β-AR signalling is vastly changed. β1-AR mRNA, and protein levels are downregulated [219,220]. The β2-AR protein level remains stable [221], but the resulting ratio of β1/β2-AR changes from 80:20 to 60:40 in the failing heart [218]. In HF, increased myocardial β3-AR gene expression and enhanced functional responses have been observed in humans [222,223], as well as in multiple animal models [222,224,225,226]. Moreover, G(i)-proteins involved in β3-AR signalling are overexpressed in HF [227,228,229]. A patent review of β3-AR ligands suggests that selective β3-AR agonists may be beneficial in the early stages of HF, whereas selective antagonists or inverse agonists may be more appropriate in the advanced stages of the disease [230]. That aligns with the reported protective role of β3-AR upregulation in acute HF, when a β3-AR-mediated decrease in contractility can be advantageous, but which later becomes maladaptive in chronic HF. In the following section of this review, we focus on ligands targeting β3-AR and their roles in HF, highlighting proposed mechanisms in various animal models and clinical trials.

#### 4.4.1. Role of β3-AR in Cardiac Remodelling and Haemodynamic Changes

The cardioprotective role of NO in acute HF is reported in several studies [231,232,233,234,235]. The β3-AR/NO/cGMP signalling cascade protects ventricular muscle from overstimulation [14,236], pressure overload, and pressure overload-induced remodelling, along with oxidative stress, as mice lacking β3-AR present exacerbated oxidative stress and adverse cardiac remodelling [237]. These effects are mediated by NO production (Figure 4), as nNOS inhibition or knockout mitigates the cardioprotective effects of BRL37344 in the pressure overload mouse HF model [238].

Cardiac-specific expression of β3-AR reduces oxidative stress, leading to a decrease in the connective tissue growth factor (CTGF) level, a paracrine factor responsible for fibroblast activation and fibrosis. These antioxidant and antifibrotic effects are abolished by either nNOS inhibition or gene deletion [239]. Moreover, overexpression of β3-AR in vivo (transgenic mice with cardiac human β3-AR) and in vitro (rat neonatal cardiac myocytes with adenoviral expression of human β3-AR) mitigates a deleterious response to angiotensin II (in vivo) and isoproterenol (in vivo and in vitro), as well as phenylephrine and endothelin 1 (both in vitro) [15,240]. As a result, it reduces cardiac hypertrophy and interstitial fibrosis without affecting myocyte death or apoptosis in these models. These effects depend on nNOS activation and subsequent downregulation of remodelling factors: nuclear factor of activated T-cells (NFAT) via cGMP-activated protein kinase G (PKG) and transforming growth factor-β1 (TGFβ1) [240], along with its intracellular signal transducers—phosphorylated Smad2/3 proteins [241]. Apparently, β3-AR-activated PKG might also activate the JNK pathway, thus upregulating transforming growth factor β (TGFβ). It is reported for primary neonatal cardiomyocytes overexpressing β3-AR and treated with BRL37344. The effect is attenuated by SR59230A [242]. This role of β3-AR might be significant because TGFβ is important in the progression of cardiac remodelling in HF, and the inhibition of JNK1/2 by chrysophanol in a rat acute HF model is cardioprotective [243]. Nonetheless, the above results suggest the benefits of chronic β3-AR stimulation, especially in Heart Failure with Preserved Ejection Fraction (HFpEF) [241]. An increased norepinephrine level during HF activates β3-AR, which induces the browning of adipose tissue. Cardiac fibroblasts and myocytes are affected by exosomes containing mRNA or proteins derived from brown adipocytes. In the mouse in vivo model, Ang-II-induced cardiac remodelling is alleviated by exosomes from mirabegron-treated adipocytes and worsened by exosomes from β3-AR-deficient adipocytes, as the latter ones contain iNOS, exacerbating oxidative stress. These results show interesting cross-talk between adipocytes and hearts mediated by iNOS-containing exosomes that could be influenced by mirabegron [244].

Interestingly, in traditional Chinese medicine, aucubin, part of *Eucommia ulmoides* Oliv, was used as a medication. It showed antioxidative, neuroprotective, anti-inflammatory, and antifibrotic effects, as well as suppression of pressure overload-induced cardiac hypertrophy in the mouse HF model. That mechanism was proven to be connected to β3-AR-mediated NO synthesis; thus, the medication is a potentially useful agent in the treatment of cardiac remodelling in HF [245]. Nebivolol in patients with HF classified as NYHA I–II (New York Heart Association, class I and II) has been shown to increase the ejection fraction and reduce the heart rate, left ventricular end-diastolic pressure, pulmonary capillary pressure, and peripheral vascular resistance [246]. In the SENIORS trial, nebivolol significantly lowered HF-related hospitalisations in elderly patients. The results were most likely due to β3-AR activation, NO release, and subsequent vasodilation, thereby contributing to the therapeutic benefit [247,248].

Some studies, particularly those examining chronic β3-AR activation, report findings that contradict the proposed cardioprotective role of β3-AR-mediated NOS activation in heart failure. Chronic administration of SR59230A, a selective β3-AR antagonist, to rats with isoproterenol-induced HF attenuates, but does not completely reverse, cardiac remodelling. The treatment leads to reductions in heart and left ventricular mass, fibrosis, and necrosis, as well as decreased myocyte edema and degeneration [214]. These outcomes might result from the inhibition of eNOS, which alleviates negative inotropism and increases ejection fraction, as well as decreases intracardiac pressure (Figure 4) [214]. Similarly, in the canine coronary microembolization heart failure model, another β3-AR antagonist demonstrates favourable haemodynamic effects, thereby contributing to disease amelioration [249]. Interestingly, Napp and colleagues suggest that in a failing myocardium, β3-AR stimulation deactivates, rather than activates, eNOS. Nonetheless, in the left ventricular tissue of a failing human heart, BRL37344 still decreases contractility, but this effect is absent in the presence of a non-selective NOS blocker—L-NMA (L-N*^ω^*-Methylarginine) [250]. The benefits of indirect β3-AR inhibition may partly explain why carvedilol is more effective than metoprolol in improving haemodynamics and cardiac remodelling in HF, as demonstrated in a rat model [251], as well as the improved survival observed in patients with heart failure in the Carvedilol Or Metoprolol European Trial (COMET) [252].

#### 4.4.2. Other β3-AR Effects in HF

##### Na^+^/K^+^-ATPase Stimulation in HF

During heart failure, a significantly elevated intracellular sodium level impairs cardiac performance; therefore, stimulation of the Na^+^/K^+^-ATPase appears to be beneficial [253]. β3-AR stimulation increases Na^+^/K^+^-ATPase activity, most likely due to antioxidant effects [254,255] (Figure 4). It has been demonstrated for BRL37344 and CL316243 in rabbit cardiomyocytes, resulting in a cardioprotective effect, most likely due to the export of excess sodium from cells [256]. In another study, reduction in sodium content and late sodium current were proposed to be especially helpful in HFpEF due to improved relaxation and direct regulation of excitation–contraction coupling [257]. In another rat model study, BRL37344-activated β3-AR reversed noradrenaline-mediated downregulation of the Na^+^/K^+^-ATPase α2-subunit, thereby alleviating haemodynamic changes and myocardial remodelling [258].

##### Sepsis-Related HF

In a septic human heart, there is an upregulation of β3-AR, compared to a non-septic myocardial tissue [259]. In a study conducted on a mouse sepsis-related myocardial dysfunction model, researchers checked the role of β3-AR using LPS-induced endotoxemia. Three groups were treated with CL316243, SR59230A, or normal saline. Results suggest that β3-AR agonists deplete myocardial energy production, most likely due to impaired glycolysis and fatty acid oxidation, which is mediated by CPT1 downregulation (Figure 4). On the other hand, the authors concluded that the blockade of β3-AR suppresses sepsis-related HF, regulates cardiac metabolism, and enhances prognosis by restoring mitochondrial biogenesis. These improvements are likely due to reduced iNOS, which downregulates glucose and fatty acid metabolism [260]. However, another study demonstrated that cardiac-specific overexpression of the eNOS gene exerted a cardioprotective effect in the mouse model of induced septic shock, likely due to increased NO production. Increased NO levels attenuated endotoxin-induced reactive oxygen species and increased myofilament sensitivity to calcium and total reticulum calcium load, which resulted in reduced myocardial dysfunction and mortality [261].

##### Autoantibodies Against β3-AR in Heart Failure

Autoantibodies against β3-AR show agonist-like effects [262]. Interestingly, serum levels of anti-β3 antibodies were elevated in patients with chronic heart failure, with 40.8% of patients being seropositive—a significantly higher proportion than in the non-HF group [262]. In a study of rats subjected to abdominal aortic banding, the group injected with anti-β3 antibodies exhibited higher left ventricular systolic pressure and lower left ventricular end-diastolic pressure (LVEDP) compared to the non-injected group. These results demonstrate a substantial cardioprotective role of anti-β3 antibodies [262]. Moreover, another study also reported increased serum levels of anti-β3 antibodies in HF patients, with the authors suggesting that higher autoantibody titres may exert negative inotropic and chronotropic effects, potentially contributing to the pathophysiology of heart failure [263]. Additionally, cardiac anti-β3 antibody seropositivity has been associated with a higher prevalence of AF and chronic obstructive pulmonary disease, indicating a possible role in other pathologies [264].

##### Role of β3-AR Agonism in Chemotherapy-Induced HF

High levels of doxorubicin in chemotherapy decrease β3-AR expression levels in myocytes [265]. In a rat model of doxorubicin-induced HF, mirabegron alone, or in combination with losartan, improved haemodynamic function. Moreover, the therapy reduced cardiac fibrosis, the expression of both interleukin 1 and 6, as well as Smad2 and 3. Interestingly, these cardioprotective effects appear to be independent of the β3-AR/eNOS-mediated pathway. The study shows that mirabegron, alone or in combination with losartan, may be a promising agent in preventing chemotherapy-induced HF [266].

##### β3-AR Upregulation in Diabetic Hearts

Upregulation of β3-AR in diabetic hearts has been reported by several groups [267,268]. In rat hearts with streptozotocin-induced diabetes (which mimics type 1/3 diabetes), β3-AR is increased, whereas β1-AR is downregulated. β2-AR is either upregulated or unchanged [269,270]. Interestingly, blocking β1-AR with metoprolol leads to the upregulation of all β-AR subtypes, with a more pronounced effect observed in the diabetic group. The treatment increases phosphorylation of Akt kinase and activation of the PI3K/Akt signalling pathway associated with activation of the G(i) protein coupled with either β2 or β3-AR. On the other hand, PKA activity is decreased, indicating increased β3-AR and decreased β2-AR activity. Prolonged activation of the β1-AR causes a switch from PKA to calcium/calmodulin-dependent protein kinase-II (CAMK-II)-dependent signalling, leading to apoptosis and pathological hypertrophy [269], whereas the PI3K/Akt pathway is cardioprotective (Figure 4) as it increases eNOS activity and deactivates proapoptotic protein forkhead transcription factor-3A (FOXO-3A). However, in this experimental model, both iNOS and eNOS are decreased [269]. In rats with streptozotocin-induced diabetes, exercise training normalizes the level of β3-AR and downregulates β2-AR, mitigating the response to isoproterenol. Nevertheless, in mice on a high-fat and sucrose diet, which mimics obesity and type 2 diabetes, the heart β3-AR expression is diminished [213], showing variable influence of different diabetes models on β3-AR.

##### Role in Improving Exercise Ability of HF-Mediated Through β3-AR

A study on a dog model with pacing-induced HF, which included the usage of the β3-AR antagonist, L748337, has shown that blockade of the receptor improves LV diastolic filling, increases LV contractility, LV arterial coupling, and mechanical efficiency, resulting in improved exercise performance [271]. A previous study of the same authors showed that stimulation of β3-AR by endogenous catecholamines contributes to the worsening of LV contraction and relaxation [225]. Another study showed that moderate aerobic exercise in a mouse model may improve recovery from pathological remodelling in failing hearts and systolic function and decrease cardiac fibrosis and hypertrophy. These effects are associated with the β3-AR pathway, including nNOS-mediated NO production and reduced oxidative stress [272].

##### Cardiorenal Syndrome

Interestingly, in a rat model of HF with cardiorenal syndrome, hypoxia of the adrenal cortex increased β3-AR expression in the zona glomerulosa. This led to cholesterol ester hydrolysis via phosphorylation of hormone-sensitive lipase (HSL) through ERK activation, ultimately resulting in elevated aldosterone production. These effects occurred independently of the renin–angiotensin system, suggesting that β3-AR function in organs beyond the heart may be relevant for heart failure treatment [273].

#### 4.4.3. Clinical Trials Involving β3-Adrenergic Receptor Stimulation in Heart Failure Patients

Considering promising results in preclinical studies, some clinical trials including the β3-AR agonist, mirabegron, were performed on HF patients. Interestingly, these trials failed to demonstrate significant benefits, contradicting the positive outcomes observed in preclinical studies [14].

In the β3-adrenergic receptor, the Left Ventricular Hypertrophy (β3-LVH) trial patients with LV hypertrophy, and those with or without HF symptoms, were given 50 mg of mirabegron per day or placebo for a duration of 12 months. After that time, LV mass and diastolic function were measured, but no significant differences were observed between the mirabegron and placebo groups [274].

The β3 Agonist Treatment in Heart Failure (BEAT-HF) clinical trial included HFrEF patients with LVEF < 40% and NYHA II-III. Patients were given 300 mg of mirabegron per day or a placebo for a duration of 6 months. Results showed no change in LVEF, volumetric parameters, physical capacity, and electrocardiogram; however, treatment was generally well tolerated [275,276].

The β3 Agonist Treatment in Heart Failure (BEAT-HF-II) trial was conducted on HFrEF patients with NYHA III–IV, LVEF < 35%, and increased N-terminal pro b-type natriuretic peptide (NT-proBNP) levels. Patients were given 300 mg of mirabegron per day or a placebo for a duration of 1 week. Mirabegron therapy improved cardiac performance by increasing the cardiac index and reducing pulmonary vascular resistance (PVR). There was no significant change in blood pressure, heart rate, and systemic vascular resistance between the mirabegron- and placebo-treated groups. This result suggests that mirabegron therapy may be useful in patients with worsening or terminal HF. BEAT-HF-II is the only clinical trial with promising results. However, considering that the treated group consisted of only 22 patients, another clinical trial with a larger sample size should be conducted [144].

## 5. Conclusions and Future Perspectives

Numerous studies suggest the potential benefits of pharmacological targeting of β3-ARs in various cardiovascular diseases. However, clinically proven cardiovascular effects have been demonstrated only for nebivolol and carvedilol, in which β3-AR activation represents a secondary mechanism. Moreover, some of these effects remain only hypothetically associated with β3-AR. Receptor activation can elicit diverse responses depending on the experimental model, particularly in relation to inflammation, fibrosis, and ion channel function. The variability in molecular responses and the phenomenon of ligand-directed agonism should be carefully considered in both study design and data interpretation. Despite a strong theoretical background, most findings come from animal models, which may not fully reflect human β3-AR physiology. Moreover, some of the proposed mechanisms are based on single studies and require further confirmation. Therefore, this paper highlights the urgent need for mechanistic studies in humans and large-scale clinical trials, as β3-AR remains a promising but underexplored target in cardiovascular therapy.

## Figures and Tables

**Figure 1 ijms-26-11844-f001:**
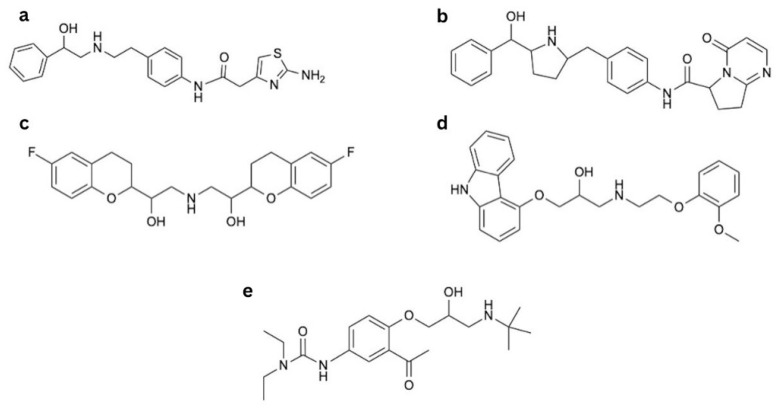
**Drugs affecting β3-AR:** (**a**) mirabegron, (**b**) vibegron, (**c**) nebivolol, (**d**) carvedilol, and (**e**) celiprolol.

**Figure 2 ijms-26-11844-f002:**
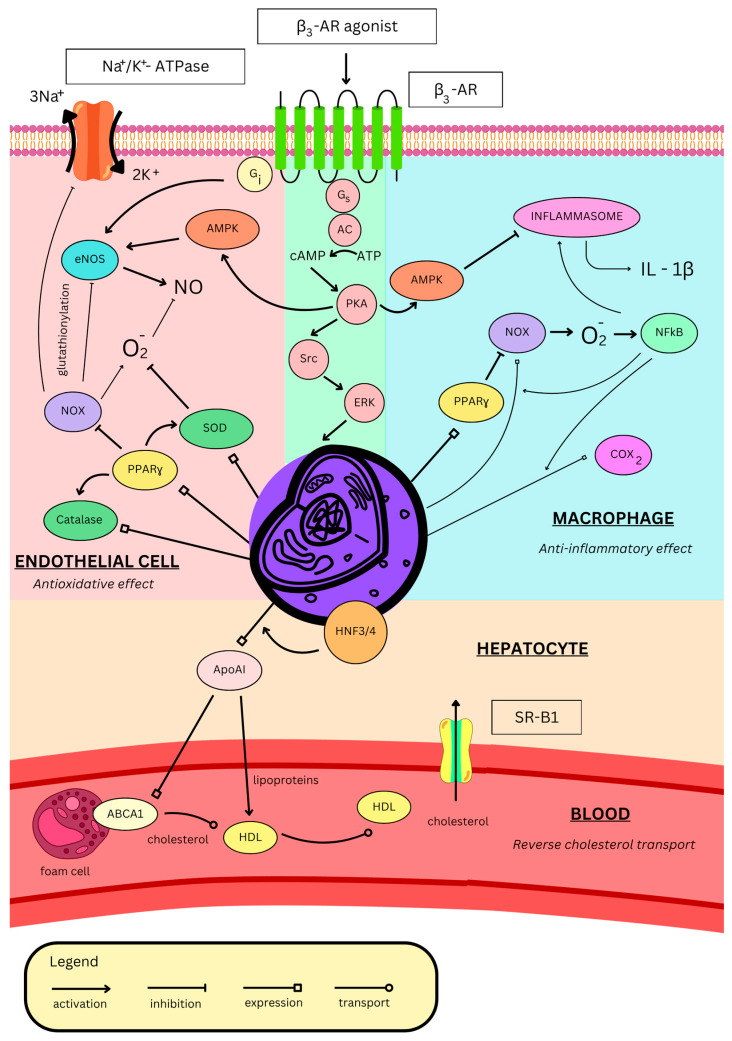
**Effects of β3-AR stimulation on blood vessels.** β_3_-AR—β3 adrenergic receptor, AMPK—adenosine monophosphate-activated protein kinase, eNOS—endothelial nitric monoxide synthase, NOX—reduced nicotinamide adenine dinucleotide phosphate oxidase, PPARγ—peroxisome proliferator-activated receptor γ, SOD—superoxide dismutase, AC—adenylyl cyclase, ATP—adenosine triphosphate, cAMP—cyclic adenosine monophosphate, PKA—protein kinase A, Src—proto-oncogene tyrosine-protein kinase Src, ERK—extracellular signal-regulated kinases, IL-1β—interleukin 1β, COX_2_—cyclooxygenase-2, NFκB—nuclear factor κ-light-chain-enhancer of activated B cells, HNF-3/4—hepatocyte nuclear factors 3 and 4, ApoAI—apolipoprotein AI, ABCA1—ATP-binding cassette A1, HDL—high-density lipoprotein, and SR-B1—scavenger receptor B1.

**Figure 3 ijms-26-11844-f003:**
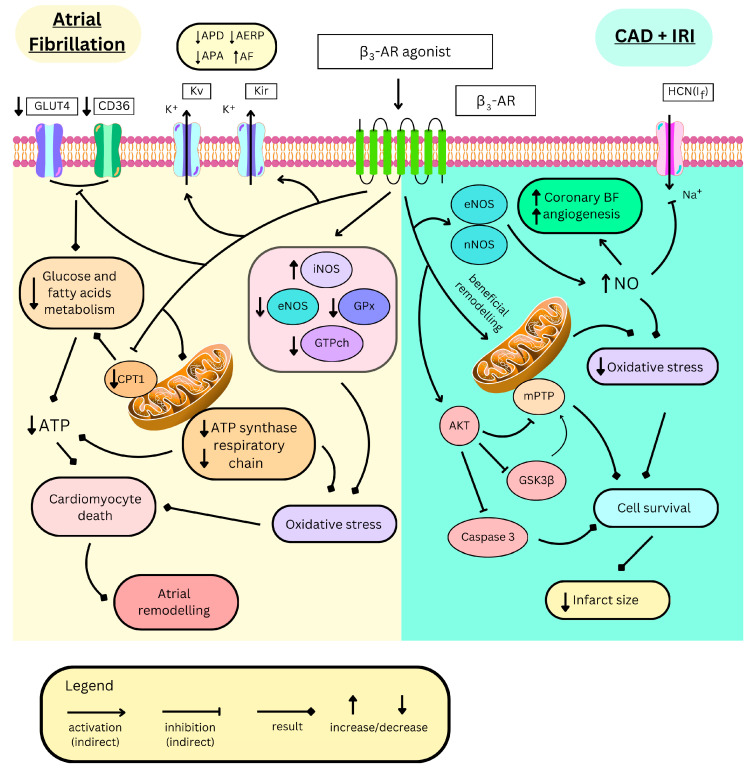
**Distinct mechanisms of β3-AR mediating deleterious effects in atrial fibrillation and beneficial effects in coronary artery disease (CAD) and ischemia–reperfusion injury (IRI).** β_3_-AR—β3 adrenergic receptor, GLUT4—glucose transporter 4, CD36—cluster of differentiation 36/fatty acid translocase, Kv—voltage-gated potassium channels, Kir—inwardly rectifying potassium channels, APD—action potential duration, AERP—atrial effective refractory period, APA—action potential amplitude, AF—atrial fibrillation, eNOS/iNOS/nNOS endothelial/inducible/neuronal nitric monoxide synthase, GPx—glutathione peroxidase, GTPch—guanosine triphosphate cyclohydrolase, CPT1—carnitine palmitoyltransferase-1, ATP—adenosine triphosphate, BF—blood flow, HCN(I_f_)—hyperpolarization activated cyclic nucleotide-gated channel, AKT—protein kinase B, NO—nitric monoxide, mPTP—mitochondrial permeability transition pore, and GSK3β—glycogen synthase kinase-3beta.

**Figure 4 ijms-26-11844-f004:**
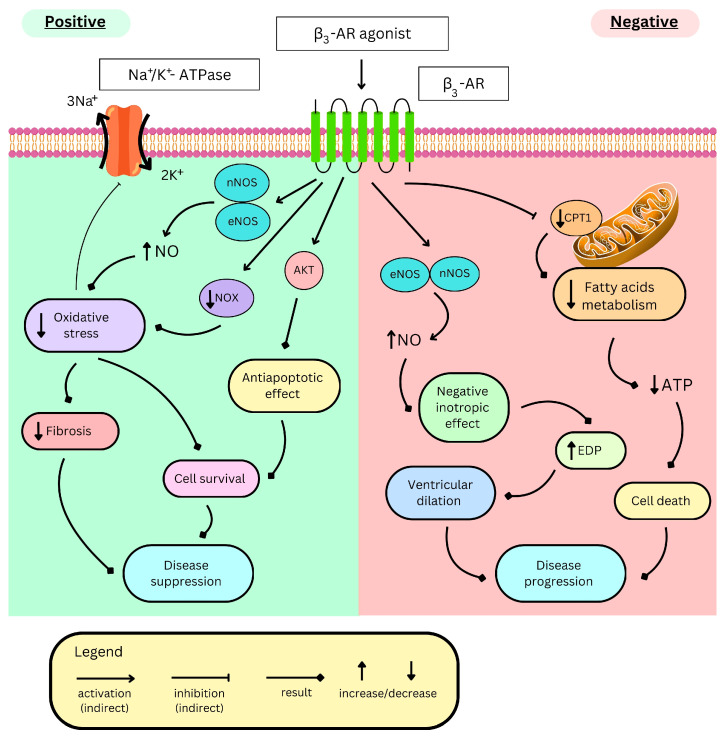
**Opposite effects of β3-AR stimulation in heart failure.** β_3_-AR—β3 adrenergic receptor, eNOS/nNOS—endothelial/neuronal nitric monoxide synthase, NO—nitric monoxide, NOX—reduced nicotinamide adenine dinucleotide phosphate oxidase, AKT—protein kinase B, CPT1—carnitine palmitoyltransferase-1, ATP—adenosine triphosphate, and EDP—end-diastolic pressure.

## Data Availability

No new data were created or analyzed in this study. Data sharing is not applicable to this article.

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
