# Peer review of "Therapeutic Potential of the β3-Adrenergic Receptor and Its Ligands in Cardiovascular Diseases"

_ijms, 2025, doi:10.3390/ijms262411844_

Round 1

Reviewer 1 Report

Comments and Suggestions for Authors

1) In the general pharmacology of β3-AR section (section 2.1), the authors need to add a brief discussion of this receptor`s affinities for the endogenous catecholamines, i.e., its natural agonists, especially given that β3-AR has much higher affinity for norepinephrine than epinephrine, which is in contrast with the other two β-AR subtypes. 

2) Lines 855-863: Since β3-AR appears to be the only functional cardiac β-AR subtype in chronic human HF, thanks to its resistance to desensitization, but still, it can increase neither cAMP (PMID: 37403791), nor contractility (PMID: 8755628), in response to norepinephrine that is elevated in chronic human HF (PMID: 3976704), the authors need to discuss in detail the physiological implications of β3-AR upregulation (Ref. 212) in chronic HF, vis a vis β3-AR acting as an adaptive (or maladaptive) mechanism in human HF.

3) Combining sections 4.4.2-4.4.7 into one, e.g., under a title: "Other β3-AR effects in HF" might make more sense, given that all these sections are short in length and HF-related.

4) Similarly, sections 4.4.8-4.4.9.3 could be combined into one, under a title: "Clinical β3-AR-related effects in HF".

5) Lines 76-78: As elegantly concluded by Ref. 19, the effects of agonists and antagonists on β3-AR expression are far from settled; they are still under investigation and definitely cell/tissue type-dependent. 

6) The title contains a redundancy: it mentions both "therapeutic potential" and "pharmacotherapy". Perhaps the word "therapeutic" before the word "potential" should be deleted.

7) Line 34: There are actually three α1 and three α2 subtypes.

8) Lines 35-36: These are actually G protein families and types. Every G protein consists of three subunits: α, β, and γ.

9) Lines 1001-1002: "Prolonged activation of the PKA pathway leads to 1001 apoptosis and pathological hypertrophy". Where is the citation(s) reporting this?

10) Line 33: Dopamine is also an endogenous catecholamine but, to my knowledge, does not activate the β3-AR.

11) Line 151: Isoproterenol is chemically a catecholamine, as well.

Author Response

Comments 1: “In the general pharmacology of β3-AR section (section 2.1), the authors need to add a brief discussion of this receptor`s affinities for the endogenous catecholamines, i.e., its natural agonists, especially given that β3-AR has much higher affinity for norepinephrine than epinephrine, which is in contrast with the other two β-AR subtypes.”
Response 1: Thank you for your suggestion. We have added a brief discussion upon the subject. Lines 113-117, marked yellow

Comments 2: “Lines 855-863: Since β3-AR appears to be the only functional cardiac β-AR subtype in chronic human HF, thanks to its resistance to desensitization, but still, it can increase neither cAMP (PMID: 37403791), nor contractility (PMID: 8755628), in response to norepinephrine that is elevated in chronic human HF (PMID: 3976704), the authors need to discuss in detail the physiological implications of β3-AR upregulation (Ref. 212) in chronic HF, vis a vis β3-AR acting as an adaptive (or maladaptive) mechanism in human HF”
Response 2: We appreciate your insightful comment regarding the physiological implications of β3-AR upregulation in the chronic human heart failure (HF).  We added a sentence in lines: 896-903. We would like to note that a detailed discussion upon the role of β3-AR in chronic HF had been included in  later sections of the manuscript.

Comments 3 and 4: “Combining sections 4.4.2-4.4.7 into one, e.g., under a title: <Other β3-AR effects in HF> might make more sense, given that all these sections are short in length and HF-related.”; “Similarly, sections 4.4.8-4.4.9.3 could be combined into one, under a title: <Clinical β3-AR-related effects in HF>.”
Response 3 and 4: Thank you for pointing this out. We agree with this comment. Therefore we have changed aforementioned section division. We have created section “4.4.2. Other β3-AR effects in HF” where we have included earlier sections 4.4.2-4.4.8. We have left them divided into subsections because we presume that this makes the manuscript more legible. Those subsections describe roles of β3-AR in very different settings of HF therefore combining them together could provide a chaotic result. Furthermore, we have combined together sections 4.4.9.-4.4.9.3. under one section: “4.4.3. Clinical trials involving β3-adrenergic receptor stimulation in heart failure patients”. In our opinion the text from previous section “4.4.7. Cardio-renal syndrome” is not suitable for section with clinical trials as it describes research performed on a rat model. Therefore we have added it to the section “4.4.2. Other β3-AR effects in HF”.

Comments 5: “Lines 76-78: As elegantly concluded by Ref. 19, the effects of agonists and antagonists on β3-AR expression are far from settled; they are still under investigation and definitely cell/tissue type-dependent.”
Response 5: Thank you for your insightful comment with whom we fully agree. We have revised that fragment so it is more accurate according to the current state of knowledge, lines: 76-82, marked yellow

Comments 6: “The title contains a redundancy: it mentions both "therapeutic potential" and "pharmacotherapy". Perhaps the word "therapeutic" before the word "potential" should be deleted.”
Response 6: Thank you for pointing this out. We fully agree with your observation. Therefore, we have changed the title to: “Therapeutic Potential of the β3-Adrenergic Receptor and Its Ligands in Cardiovascular Diseases”. In this title there is no redundancy and in our opinion it is more comprehensive.

Comments 7: “Line 34: There are actually three α1 and three α2 subtypes.”
Response 7: Agree. We have, accordingly, revised this fragment to emphasize the subdivision – lines 33-35, marked yellow

Comments 8: “Lines 35-36: These are actually G protein families and types. Every G protein consists of three subunits: α, β, and γ.”
Response 8: Thank you for pointing this out. We agree with your comment. We have revised this fragment so it is now unanimous with the well-established knowledge. Line 36-38, marked yellow

Comments 9: “Lines 1001-1002: <Prolonged activation of the PKA pathway leads to 1001 apoptosis and pathological hypertrophy>. Where is the citation(s) reporting this?”
Response 9: Thank you for pointing this out. To be more precise we have changed that fragment – lines 1035-1037 and we have added ref. 271 (previously ref. 262) to this place – marked yellow.

Comments 10: “Line 33: Dopamine is also an endogenous catecholamine but, to my knowledge, does not activate the β3-AR.”
Response 10: Agree. We have, accordingly, changed the word “such as” to “primarily” in line 33 to emphasize that according to current knowledge epinephrine and norepinephrine are main endogenous catecholamines activating β3-AR.

Comments 11: “Line 151: Isoproterenol is chemically a catecholamine, as well.”
Response 11: We agree  and we have deleted the phrase “and isoproterenol” so it will not be further misleading. “Its activation requires higher concentrations of catecholamines [36]line 157, marked yellow

Reviewer 2 Report

Comments and Suggestions for Authors

This is a very comprehensive and interesting review that provides a thorough focus on β3-adrenergic receptor agonists and their pharmacodynamics with respect to receptor binding and activation. The authors skillfully describe the mechanisms of action mediated by β3-AR in blood vessels and the heart, covering both physiological and pathological conditions. They particularly emphasize the therapeutic potential in arterial, pulmonary, and portal hypertension, as well as cardiac arrhythmias, ischemia, and heart failure, considering both receptor agonism and antagonism.

Due to the large amount of information and numerous abbreviations, the text can be somewhat challenging to follow at times. Nevertheless, this review represents an excellent and valuable contribution, offering a detailed and insightful synthesis of the current knowledge on β3-AR pharmacology and its cardiovascular implications. However, I suggest minor corrections that will improve scientific impact of this manuscript, as stated below:

  1. In line 45, the manuscript states thatβ3-AR is expressed in many tissues. For clarity and completeness, please specify the tissues where β3-adrenergic receptors are expressed, as this will strengthen the contextual understanding of their functional role.
  2. In lines 53, 54 you state:“On the other hand, nitric oxide (NO) produced by endocardial cells decreases cardiac contractility [11, 12].”
    I suggest revising “nitric oxide” to “nitric monoxide” if you are referring specifically to NO, since the term “nitric oxide” may be interpreted more broadly to include other oxides of nitrogen (e.g., N₂O, NO₂), which could lead to ambiguity.
  3. In the final paragraph of the Introduction, please indicate whether the review considered only studies that were fully accessible and written in English.
  4. Before or after discussing the individual drugs in section 4 β3-AR agonists, it would be helpful to briefly comment on the pharmacophore of these compounds. Additionally, a comparative overview of their chemical structures could help highlight subtle differences that influence binding affinity and selectivity.
  5. In the section describing individual drugs, it would be beneficial to provide a more detailed discussion of their chemical structures and to explain how these structural features influence binding to the target receptors.
  6. I would suggest separating Arterial Hypertension and Preeclampsia into two distinct subsections from 3.3.1. Arterial Hypertension.
  7. Before introducing the subsections in section 4 Role of β3-AR in the Heart, it would be helpful to provide a brief introductory paragraph explaining the significance of this section and everything that follows within it.
  8. Considering the limited available information, no need for further subdivision and fragmentation of subsection 4.4.9. Please keeping it as a single subsection.
  9. Please check the formatting of all headings and subheadings throughout the manuscript and uniform them according to journal’s instructions for authors. I noticed that sections 3.1, 3.2, and 3.3 do not follow the same style.
  10. In the manuscript, Vibegron is referred to as Vibegron (Figure 2.2). This should be corrected to Vibegron (Figure 1.2).
  11. To reduce the number if abbreviations, I suggest introducing them only for terms that appear multiple times through the text.

Author Response

Comments 1: “In line 45, the manuscript states thatβ3-AR is expressed in many tissues. For clarity and completeness, please specify the tissues where β3-adrenergic receptors are expressed, as this will strengthen the contextual understanding of their functional role.”
Response 1: Changed: “β3-AR is expressed in many tissues, including adipose tissue, the urinary bladder, the myocardium and the arterial endothelium, and plays an important role in regulating various body functions. In the adipose tissue, β3-AR was initially found to mediate lipolysis. It stimulates the expression of uncoupling protein 1 (UCP1) which plays a role in thermogenesis. In the urinary bladder it relaxes the detrusor muscle. In the myocardium and the arterial endothelium its agonists may have a potential role in the treatment of heart failure (HF) and other cardiovascular pathologies [8-10]”.

Comments 2: “In lines 53, 54 you state:<On the other hand, nitric oxide (NO) produced by endocardial cells decreases cardiac contractility [11, 12].>
I suggest revising “nitric oxide” to <nitric monoxide> if you are referring specifically to NO, since the term “nitric oxide” may be interpreted more broadly to include other oxides of nitrogen (e.g., N₂O, NO₂), which could lead to ambiguity.”
Response 2: Thank you for pointing this out. We have followed your suggestions and changed the term “nitric oxide” to “nitric monoxide” in that place where it first appears in the manuscript as in the further sections the well-established abbreviation “NO” is used.

Comments 3: “In the final paragraph of the Introduction, please indicate whether the review considered only studies that were fully accessible and written in English.”
Response 3: Agree. We have included that information. Line 94-95: “Only studies fully accessible and written in English were included.”

Comments 4 and 5: “Before or after discussing the individual drugs in section 4 β3-AR agonists, it would be helpful to briefly comment on the pharmacophore of these compounds. Additionally, a comparative overview of their chemical structures could help highlight subtle differences that influence binding affinity and selectivity.”; “In the section describing individual drugs, it would be beneficial to provide a more detailed discussion of their chemical structures and to explain how these structural features influence binding to the target receptors.”
Response 4 and 5: Thank you for pointing this out. We agree with this comment. Therefore we have added an explanatory paragraph “2.4.1. Structure-activity relationship of β3-AR agonists” (lines 186-193) where we discuss the pharmacophore of β3-AR agonists in the context of their structure-activity relationship. Additionally, to each description of an individual drug we have provided a brief discussion upon their chemical structure regarding binding to the target receptors(marked yellow).

Comments 6: “I would suggest separating Arterial Hypertension and Preeclampsia into two distinct subsections from 3.3.1. Arterial Hypertension.”
Response 6: Agree. We have separated the paragraph about preeclampsia into another subsection: “3.3.3. Preeclampsia”.

Comments 7: “Before introducing the subsections in section 4 Role of β3-AR in the Heart, it would be helpful to provide a brief introductory paragraph explaining the significance of this section and everything that follows within it”
Response 7: We fully agree with your suggestion. Therefore we have added the following fragment in this section (lines 589-595) – marked yellow

Comments 8: “Considering the limited available information, no need for further subdivision and fragmentation of subsection 4.4.9. Please keeping it as a single subsection.”
Response 8: Agree. We have combined together sections 4.4.9.-4.4.9.3. under one section: “4.4.3. Clinical trials involving β3-adrenergic receptor stimulation in heart failure patients”.

Comments 9: Please check the formatting of all headings and subheadings throughout the manuscript and uniform them according to journal’s instructions for authors. I noticed that sections 3.1, 3.2, and 3.3 do not follow the same style.
Response 9: Thank you for pointing this out. We have checked whether all sections of the manuscript follow the proper formatting according to journal’s instructions. Wherever there was a mistake, we have corrected the style.

Comments 10: “In the manuscript, Vibegron is referred to as Vibegron (Figure 2.2). This should be corrected to Vibegron (Figure 1.2).”
Response 10: Agree. We have corrected this mistake.

Comments 11: “To reduce the number if abbreviations, I suggest introducing them only for terms that appear multiple times through the text.”
Response 11: Thank you for your suggestion. We have revised all of the abbreviations.

Round 2

Reviewer 1 Report

Comments and Suggestions for Authors

No further comments.

Comments on the Quality of English Language

Quality of language is acceptable but could be improved.